# Previous infection with seasonal coronaviruses does not protect male Syrian hamsters from challenge with SARS-CoV-2

Magen E. Francis [1,2], Ethan B. Jansen [1,2], Anthony Yourkowski [1,2], Alaa Selim[1,2], Cynthia L. Swan[1], Brian K. MacPhee[1,2], Brittany Thivierge[1], Rachelle Buchanan[1], Kerry J. Lavender [2], Joseph Darbellay[1], Matthew B. Rogers[1], Jocelyne Lew[1], Volker Gerdts[1], Darryl Falzarano [1], Danuta M. Skowronski[3,4], Calvin Sjaarda [5,6] & Alyson A. Kelvin [1,2] ✉

SARS-CoV-2 variants and seasonal coronaviruses continue to cause disease and coronaviruses in the animal reservoir pose a constant spillover threat. Importantly, understanding of how previous infection may influence future exposures, especially in the context of seasonal coronaviruses and SARS-CoV-2 variants, is still limited. Here we adopted a step-wise experimental approach to examine the primary immune response and subsequent immune recall toward antigenically distinct coronaviruses using male Syrian hamsters. Hamsters were initially inoculated with seasonal coronaviruses (HCoV-NL63, HCoV-229E, or HCoV-OC43), or SARS-CoV-2 pango B lineage virus, then challenged with SARS-CoV-2 pango B lineage virus, or SARS-CoV-2 variants Beta or Omicron. Although infection with seasonal coronaviruses offered little protection against SARS-CoV-2 challenge, HCoV-NL63-infected animals had an increase of the previously elicited HCoV-NL63-specific neutralizing antibodies during challenge with SARS-CoV-2. On the other hand, primary infection with HCoV-OC43 induced distinct T cell gene signatures. Gene expression profiling indicated interferon responses and germinal center reactions to be induced during more similar primary infection-challenge combinations while signatures of increased inflammation as well as suppression of the antiviral response were observed following antigenically distant viral challenges. This work characterizes and analyzes seasonal coronaviruses effect on SARS-CoV-2 secondary infection and the findings are important for pan-coronavirus vaccine design.

Coronaviruses (family *Coronaviridae*; order *Nidovirales*) are a continual threat to human health due to the current COVID-19 (coronavirus disease from 2019) pandemic, circulation of seasonal coronaviruses, and potential for new spillover viruses[1]. Coronaviruses have positive sense, single-stranded RNA genomes and are categorized into one of four genera: alphacoronaviruses, betacoronaviruses, gammacoronaviruses, and deltacoronaviruses[2]. Human coronaviruses (HCoV) belong to either the alphacoronavirus or betacoronavirus genera suggesting these genera to be of human concern[2]. The Spike (S) protein on the viral envelope contains distinct B- and T-cell epitopes that dominate S-specific immune responses[3–5] and is responsible for host cell binding making it an attractive vaccine target. However, the S protein also represents great antigenic diversity among virus family members and is prone to antigenic drift which can lead to immune

evasion in previously exposed hosts[1]. Prior to 2002, coronaviruses represented by the seasonal coronaviruses, including alpha (HCoV-229E (229E), HCoV-NL63 (NL63)) and beta (HCoV-OC43) species, were thought to cause mild respiratory disease with limited immune memory in people[6–11]. It is estimated that adults experience 2-3 colds annually while children can have even more common cold virus events each year[12]. Additionally, several human studies suggest that immunity to common colds is short-lasting and reinfection can occur yearly[11]. While several different viruses can cause the common cold, coronaviruses are estimated to be the cause of 15-30% of common colds[13]. Therefore, although almost all people are thought to have been infected with cold viruses at some point in their lives, only 30–60% of people will have been infected with a seasonal coronavirus within the last year[14–16]. More recently betacoronaviruses have demonstrated the ability to cause severe disease associated with pneumonia, organ failure, and even death. The severe coronaviruses include Severe Acute Respiratory Syndrome coronavirus (SARS-CoV) which emerged in 2002, Middle East Respiratory Syndrome coronavirus (MERS-CoV) which emerged in 2012, and SARS-CoV-2 which emerged in 2019 and causes COVID-19[17,18]. Adaptive immune longevity after SARS-CoV-2 infection has been investigated and debated since immune durability varies greatly among people following COVID-19 recovery[19,20]. Protection from reinfection has been complicated by the emergence of variants of concern (VOCs) evading immunity, such as Beta and Omicron[21–27]. The contribution of previous existing immunity from infection with the more distant seasonal coronaviruses on COVID-19 outcomes is also not clear since human serological studies have yielded conflicting results[15,28–30]. More work is needed to understand if a previous coronavirus infection can modulate infection during a subsequent coronavirus exposure as well as the mechanisms regulating coronavirus cross-protection. This information will be important for developing pan-coronavirus vaccines that can protect against emerging and circulating coronaviruses at the same time.

To experimentally investigate immune responses over a spectrum of closely related and distantly related seasonal and severe coronavirus reinfections, we designed a study using Syrian hamsters. We first determined if previous infection with seasonal coronaviruses (NL63, OC43, and 229E) would protect against a challenge with a SARS-CoV-2 pango lineage B isolate to represent the prototypic SARS-CoV-2 virus (henceforth ancestral SARS-CoV-2 virus, or "ancestral"). The results were then compared to primary infection-challenge studies performed with ancestral SARS-CoV-2 virus and the VOCs Beta and Omicron, as these combinations represented more closely related virus exposures as well as recent exposures affecting large populations of people. Immune responses such as antibody elicitation and the induction of germinal center reaction gene profiles were investigated in pre-exposed animals during coronavirus challenge to understand cross-reactive mechanisms, both positive (protective) or negative (disease enhancing).

## Results

### NL63 and OC43 but not 229E infect Syrian hamsters
In order to investigate the impact of sequential coronavirus exposures and infection, we first required a preclinical model that would be susceptible to the coronaviruses that cause human respiratory disease which we were interested in studying. Specifically, we had the seasonal coronaviruses NL63, OC43, 229E as well as SARS-CoV-2 ancestral virus, and variants Alpha, Beta, Delta, and Omicron representing severe coronaviruses. Previously published studies of seasonal coronavirus infection mainly leveraged the mouse model; however, we chose to use Syrian hamsters for our study of human coronaviruses due the utility it has shown in SARS-CoV-2 pathogenesis studies[31–37]. Analysis of S proteins on human coronaviruses showed that the seasonal coronavirus full S protein sequences share between 25-30% identity with the SARS-CoV-2 ancestral virus, variants of concern (VOCs), and the

other seasonal coronaviruses, while 97-99% identity was observed in our analysis among the S protein of the VOCs with that of the ancestral strain (Fig. 1A). This viral sequence analysis suggested that the seasonal coronaviruses were antigenically distant to the SARS-CoV-2 ancestral strain while VOCs were more antigenically similar (Fig. 1A). To further estimate the potential susceptibility of Syrian hamsters to the coronaviruses of interest, we evaluated the similarity between the human host receptor for each virus compared to the Syrian hamster orthologues. The host receptor for NL63 and SARS-CoV-2, angiotensin-converting enzyme 2 (ACE2), had 84.5% identity between humans and Syrian hamsters[38,39]. In addition, there was 95.5% identity for 9-O-acetylated sialic acid esterase, the enzyme functionally associated with 9-O-acetylated sialic acid (the receptor for OC43 and HKU1) and 75.2% for aminopeptidase N (APN) which binds 229E[40–42]. Based on the receptor identity between human and hamster, we inferred OC43 to have the highest potential to infect Syrian hamsters and Syrian hamsters to be least susceptible to 229E.

Next, we biologically characterized the Syrian hamster's susceptibility as well as immunological and clinical outcomes of inoculation with the seasonal coronaviruses OC43, NL63, and 229E. Blood samples were taken from all animals prior to infections to confirm no animal had antibodies against any of the circulating coronaviruses. Three groups of Syrian hamsters were intranasally inoculated with OC43, NL63, or 229E at an infectious dose of $10^5$ TCID$_{50}$. Male hamsters aged 8-weeks were utilized. A total of 12 animals were used in each group (three animals sampled on days 3 and 6 post inoculation and six animals sampled on day 14 post inoculation). As a mock control, one group of hamsters was inoculated with Dulbecco's Modified Eagle Medium (DMEM) (referred to as mock). We also leveraged our previous data of SARS-CoV-2 ancestral virus inoculated hamsters at the same infectious dose to understand the clinical differences and viral dynamics of seasonal coronavirus inoculation compared to a known severe coronavirus. Clinical signs including weight loss and temperature were monitored over 14 days post inoculation (pi) and in-life nasal washes as well as necropsy samples were collected on days 1, 3, 6, and 14 for viral load and host response analysis (Fig. 1 and Supplementary Fig. 1). Weight remained close to 100% of original starting weight following inoculation with all seasonal coronaviruses and noted weight increases after day 7 pi. This trend was significantly different than the -15% weight loss observed for SARS-CoV-2 ancestral virus inoculated animals (Fig. 1B) while no differences in temperature were found among the groups (Supplementary Fig. 1A). Viral shedding and tissue viral load were assessed throughout the upper and lower respiratory tract during the time course. No live virus, determined by TCID$_{50}$ assay, was identified in the nasal washes or nasal turbinates for the animals inoculated with 229E while viral RNA was found at negligible amounts in the day post inoculation (Fig. 1Ci). However, NL63 and OC43 viral RNA and live virus were detected in both sample types in each of the virus inoculated groups (Fig. 1Cii and Ciii, respectively). Live virus was identified as early as day 1 pi, peaked on day 3 pi (approx. $10^3$ TCID$_{50}$/mL), and was still present in the nasal washes and nasal turbinates on day 6 pi for the majority of the NL63 and OC43 inoculated animals. Viral loads were lower for the seasonal coronavirus-infected hamsters than observed in SARS-CoV-2 ancestral virus inoculated Syrian hamsters which reached as high as $10^8$ TCID$_{50}$/mL[35]. Moreover, the seasonal coronavirus inoculated hamsters did not show evidence of infection in the lower respiratory tract whereas we have seen live virus in the lung and trachea of SARS-CoV-2 ancestral virus inoculated hamsters previously[35]. Viral RNA was detected in lung lobes of the NL63 inoculated animals; however, live virus was not isolated (Supplementary Fig. 1B). Extrapulmonary tissues such as the heart, spleen, and large intestines were negative for viral RNA and live virus for NL63, OC43 and 229E inoculated animals (Supplementary Fig. 1C−E). Although live virus was not identified in the lungs, histopathological analysis of lung tissue indicated immune cell infiltration and mild pathology.

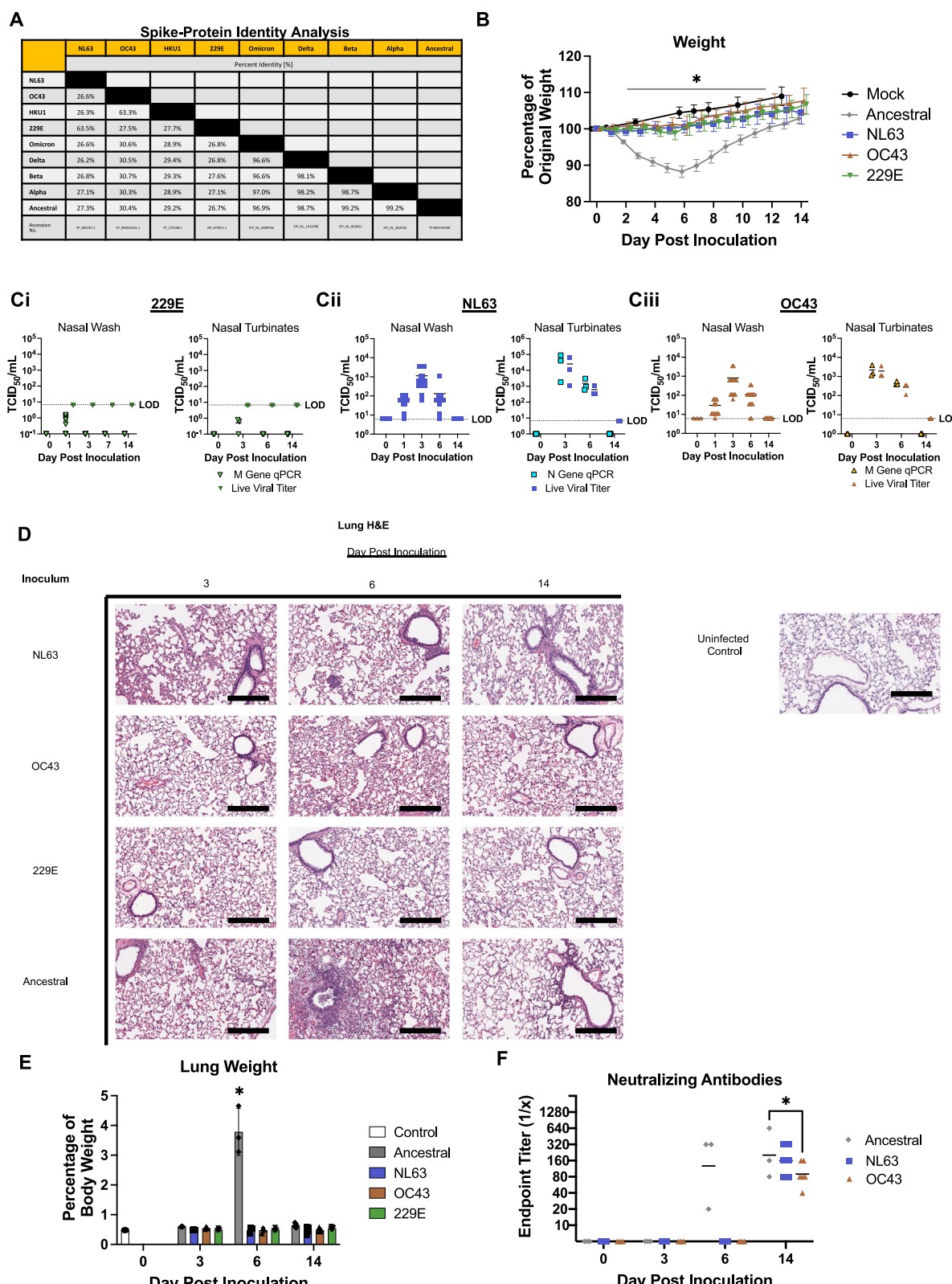

Mononuclear cells infiltrated the alveoli with associated inflammation on day 6 and 14 pi following NL63 inoculation and small clusters of macrophage-like cells were observed lasting until day 14 (Fig. 1D). The lungs of NL63 inoculated animals scored for inflammation peaking at an average of 1.3/4 on day 6 and somewhat resolving to .33/4 on average on day 14 (Supplementary Fig. 2). The OC43 inoculated animals had significant mononuclear cell infiltration (mainly macrophage-like cells and neutrophils) and hemorrhaging on day 6 pi. Edema was

also noted around a group of blood vessels in one of three animals on day 14 post OC43 inoculations (Fig. 1D). The same OC43 inoculated animal scored a 2/4 on lung inflammation with less than 25% of the lung being affected (Proportion score of 1; Supplementary Fig. 2). Not surprisingly the lungs from the 229E inoculated animals, which did not show evidence of active viral infection even in the upper respiratory tract, retained a similar architecture compared to control uninfected tissue (Fig. 1D) and scored 0/4 on lung histology (Supplementary

**Fig. 1 | Syrian hamsters are susceptible to infection with the seasonal coronaviruses NL63 and OC43, but not 229E, leading to upper respiratory viral replication and mild disease.** Identity differences among the entire proteome and the amino acid sequence of the spike protein for the coronaviruses known to infect humans (**A**). The seasonal coronaviruses NL63, OC43, and 229E were intranasally inoculated into 8-week-old Syrian hamsters at an infectious dose of $10^5$ $TCID_{50}$ and weight loss was calculated in terms of percent of original weight over 14 days. $n = 15$ animals per group (**B**). One group of hamsters were inoculated with media as control. Live virus was assessed in collected nasal washes and nasal turbinates by $TCID_{50}$ assay post inoculation for 229E (**Ci**), NL63 (**Cii**), and OC43 (**Ciii**). $n = 15$ animals per group. Collected lungs from each group post inoculation were processed for histological assessment (**D**). Lungs were formalin perfused, sectioned, mounted, and H&E stained followed by visualization and analysis by a board-certified pathologist. The entire lung was weighed at necropsy and lung gross pathology was represented as a percentage of total animal body weight. $n = 3$ control animals and 15 for all other groups (**E**). Blood was collected on days 0, 3, 6 and 14 post inoculation. Microneutralization assays were performed using the plasma samples collected from each group against their respective primary inoculating virus. $n = 15$ animals per group. Lines represent group geometric mean (**F**). At least 3 animals per timepoint were analyzed. $TCID_{50}$/mL was calculated using the Reed and Muench method. LOD = limit of detection for live viral load assays. * indicates a $p$-value less <0.05 determined by one-way ANOVA comparing hamsters on the days post inoculation to baseline (day 0) for Fig. 1E and comparing groups on the same day for all other comparisons. Data are presented as mean values with error bars indicating ±SD as appropriate. Slides were visualized and imaged using the Aperio ScanScope XT. Images were captured at 10X. Scale bars represent 300 μm. Images shown are representative of 3 animals per group, per timepoint. Source data are provided as a Source Data file.

Fig. 2). The SARS-CoV-2 ancestral virus inoculated animals displayed evidence of bronchopneumonia, significant peribronchiolar thickening, hemorrhaging, interstitial pneumonia, and mononuclear cell infiltration on day 6 pi (Fig. 1D), as we have previously reported[35]. Ancestral virus inoculated animals scored on all histology parameters, with over 50% of the lung being affected on day 6 and 14, and highest scoring in any category being on day 6 with an average score of 3 in bronchiole, corresponding to marked epithelial lesions (Supplementary Fig. 2). No significant differences were identified for lung to body weight ratio for any of the seasonal coronavirus inoculated animals while the SARS-CoV-2 ancestral virus inoculated hamsters increased their ratio 4x (Fig. 1E). Evaluation of virus neutralizing antibodies by microneutralization assay for each virus indicated that neutralizing antibodies were detected in the plasma of the respective OC43 and NL63 inoculated hamsters by day 14 pi. These titers were observed at 1:160 and 1:80, respectively, which were significantly lower than virus neutralizing antibodies against ancestral SARS-CoV-2 in the ancestral inoculated hamsters by day 14 pi (Fig. 1F). No neutralizing antibodies were detected in the 229E inoculated animals (Supplementary Fig. 3A). No cross-neutralization was detected for any of the seasonal coronavirus inoculated animals (Supplementary Fig. 3A–C) and mock inoculated animals were also negative for any seasonal coronavirus neutralizing antibodies (Supplementary Fig. 3D). This data suggests different immune dynamics regulate the adaptive immune responses for severe coronavirus infection compared to seasonal.

To understand the immune responses in the respiratory tissues of NL63 and OC43 inoculated animals, we assessed host gene expression of select immune genes via qPCR. In the nasal turbinates (Fig. 2 and Supplementary Fig. 4), both NL63 (A) and OC43 (B) inoculated animals had upregulation of type I (*IFN-β, STAT2, IRF1, IRF3, TLR3*) and type II IFN genes (*IFN-γ, IRF2, STAT1, CXCL10*). This contrasted with SARS-CoV-2 ancestral virus inoculated animals (Fig. 2C and Supplementary Fig. 4C) where *IFN-β* was downregulated and other type I IFN genes were not induced, while type II IFN gene regulation was intact, as we have previously shown[35]. All inflammatory genes assessed (*IL-6, TNF, IL-1β*) were upregulated significantly (fold change >20 compared to baseline) in the nasal turbinates (Fig. 2 and Supplementary Fig. 4) of ancestral virus inoculated animals (C) while NL63 (A) and OC43 (B) inoculated animals had minimal upregulation of *TNF* (fold change less than 10 compared to baseline) in the nasal turbinates (Fig. 2 and Supplementary Fig. 4) and no significant upregulation in the right cranial lung (Fig. 2 and Supplementary Fig. 5). Overall, there was less regulation of immune genes in the lung (Fig. 2 and Supplementary Fig. 5) compared to the nasal turbinates (Fig. 2 and Supplementary Fig. 4) of seasonally inoculated (A and B) animals. Notably the nasal turbinates were the only tissue where live virus was detected in NL63 and OC43 inoculated animals (Fig. 1Cii, iii). Taken together, we have shown distinct immune gene profiles in the seasonal coronavirus animals compared to the more pathogenic coronavirus, SARS-CoV-2 ancestral virus.

## OC43 inoculated animals had faster viral clearance at ancestral SARS-CoV-2 secondary challenge while neutralizing antibodies were back-boosted in NL63 inoculated animals with no protective benefits

As we were successful in developing seasonal coronavirus preclinical models for NL63 and OC43 using Syrian hamsters, we went on to determine the impact and mechanisms of seasonal coronavirus inoculation on a SARS-CoV-2 ancestral virus secondary challenge given the conflicting results reported in human clinical studies[15,28,29]. A total of 18 male Syrian hamsters were used in each group (three animals sampled on days 0, 3, 6, and 9 post secondary challenge and six animals sampled on day 14 post secondary challenge). Syrian hamsters were intranasally inoculated with NL63 or OC43 to establish pre-existing immunity for a subsequent heterologous secondary challenge with ancestral SARS-CoV-2 virus. These groups are designated NL63-Ancestral and OC43-Ancestral to represent the primary inoculation-challenge combination. A group of positive control hamsters were inoculated with ancestral virus for a homologous ancestral virus secondary challenge (Ancestral-Ancestral). DMEM was intranasally instilled in a fourth group to establish a non-inoculated and age-matched control group for challenge comparison (Mock-Ancestral). All animals were inoculated on day 0 with their respective viruses at $10^5$ $TCID_{50}$ to establish pre-existing immunity. Animals were then rested until day 56 post primary inoculation (ppi) when all groups were intranasally challenged with ancestral virus at $10^5$ $TCID_{50}$ and observed for 14 days post secondary challenge (psc) (Fig. 3A). Nasal wash and tissue samples were collected throughout the time course including blood samples on day 55 ppi to determine antibody titers prior to secondary challenge. As expected, weight loss of the homologous Ancestral-Ancestral animals was minimal and significantly less than the weight loss during the study day 56 ppi challenge period of the Mock-Ancestral control animals. Moreover, both NL63 and OC43 inoculated animals had weight loss signatures similar to that of the mock-inoculated animals after ancestral virus challenge where weights reached the lowest point on day 6 psc at -85% of original weight (Fig. 3B).

Viral load (vRNA and live virus) was quantified in the nasal washes and the upper and lower respiratory tract tissue post ancestral virus secondary challenge (Fig. 3C and Supplementary Fig. 6A, B). The shedding of live virus in nasal washes was similar for the mock, NL63, and OC43 inoculated groups at -$10^4$ $TCID_{50}$/mL, markedly above the homologous Ancestral-Ancestral control group which had only one animal with viral loads above the limit of detection (Fig. 3C). On day 6 psc, the viral load in the Mock-Ancestral group remained statistically increased relative to the Ancestral-Ancestral control group. Interestingly, both of the seasonal coronavirus inoculated groups had less live virus in the nasal washes on day 6 psc compared to the mock control group; however, viral loads were not statistically different than mock. On day 9 psc, both the ancestral virus and OC43 inoculated groups were no longer shedding virus in nasal washes while the NL63-Ancestral and Mock-Ancestral groups had one and two positive

## Host Gene Expression qPCR

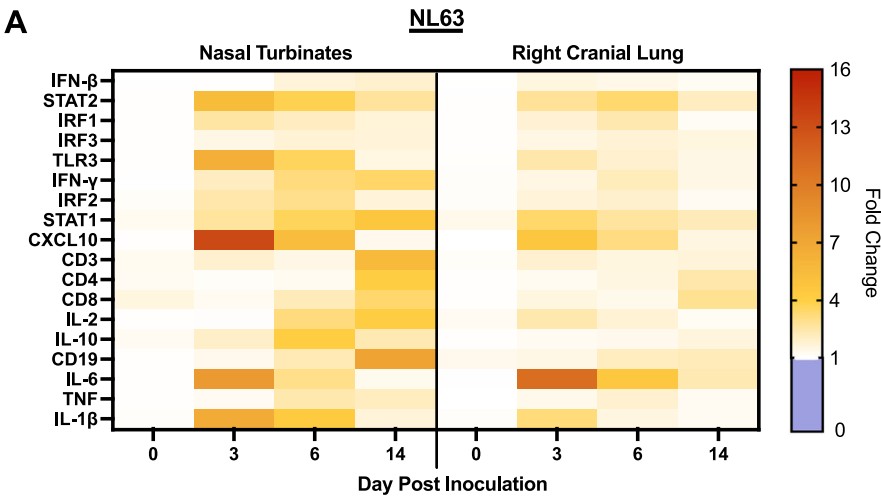

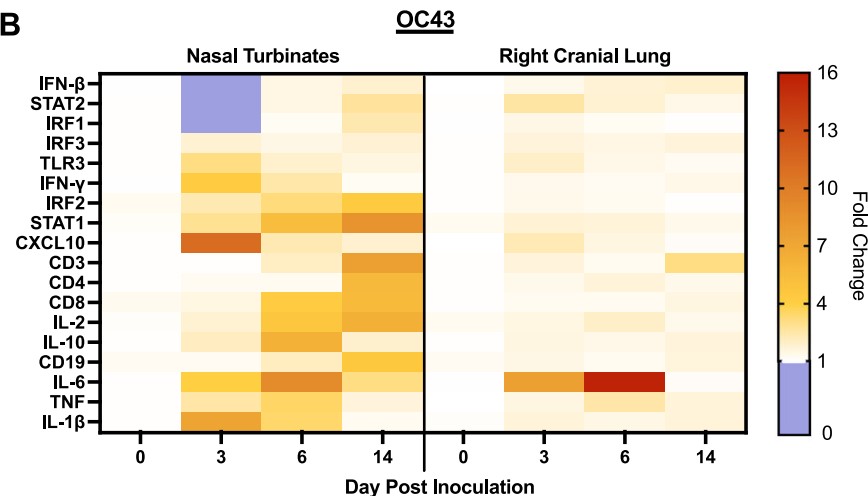

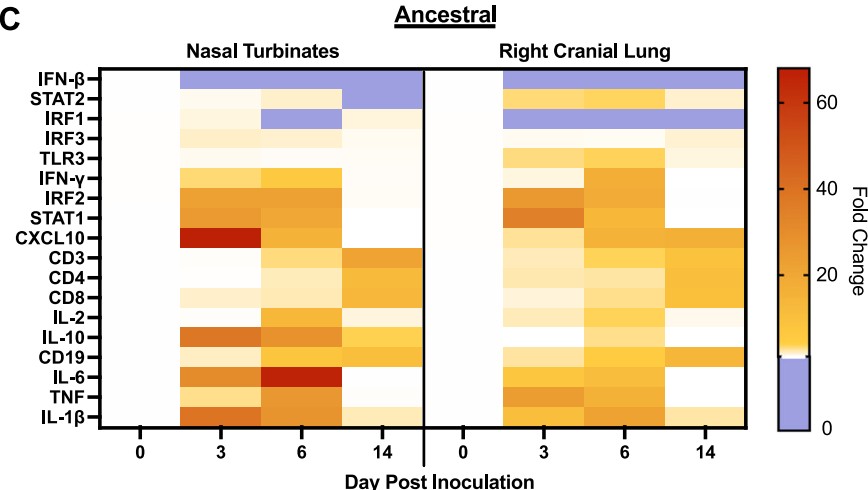

animals, respectively (Fig. 3C). No viral shedding was detected in any of the groups on day 14 psc. Tissue viral loads indicated live virus to be present following secondary ancestral virus challenge in all groups on day 3 psc in the nasal turbinates and right cranial lung, although, the homologous Ancestral-Ancestral control group had viral titers that were not detected or were significantly lower than the other groups (Fig. 3C). The NL63 inoculated group had significantly higher viral

titers in the nasal turbinates compared to the mock inoculated control group on day 3 psc. Interestingly, although not statistically significant, the OC43 inoculated group consistently had lower viral loads than NL63 and mock-inoculated groups suggesting the virus was able to infect the OC43 inoculated animals but was cleared faster (Fig. 3C). Although we previously reported live virus in the mediastinal lymph nodes of some ancestral virus infected hamsters at a younger age[35], no

**Fig. 2 | Inoculation with NL63 and OC43 led to distinct host gene expression in respiratory tissues compared to ancestral SARS-CoV-2 inoculation.** qRT-PCR was performed on RNA extracted from nasal turbinates and the right cranial lung following inoculation with NL63 (**A**), OC43 (**B**), or ancestral SARS-CoV-2 (**C**). Gene regulation was analyzed according to response type and results were displayed as a heat map according to fold regulation. Primers were designed or acquired specific to hamster genes (Table 2). Fold-change was calculated by ΔΔCt against baseline (Day 0) with *BACT* as the housekeeping gene. The legend depicts fold change in which upregulation (greater than 1) is represented ranging in from white (=1) to red. Any downregulation (<1) is represented in blue. At least three animals were analyzed for each timepoint. Statistical differences from baseline can be found in Supplementary Table 1 for NL63, Supplementary Table 2 for OC43 and Supplementary Table 3 for ancestral SARS-CoV-2. Bar graphs of this data can be found in Supplementary Fig. 4 for nasal turbinates and Supplementary Fig. 5 for right cranial lung. Source data are provided as a Source Data file.

live virus was detected in the mediastinal lymph node for any of the time points in any of the groups after secondary challenge in this current experiment (Fig. 3C).

Histopathological analysis of the lungs after secondary ancestral virus challenge showed characteristic interstitial pneumonia, hemorrhaging, and mononuclear cell infiltration into the alveolar space peaking on day 6 psc of the Mock-Ancestral group (Fig. 3D) which was similar to our previous study of ancestral SARS-CoV-2 infected younger Syrian hamsters[35]. Histology scoring correspondingly peaked on day 6 psc when there was an overall severity score of 3/4 on average and scoring of 1 or greater in all Mock-Ancestral animals for inflammation, pneumonia, and bronchioles (Supplementary Fig. 7). NL63-Ancestral animals, but not OC43-Ancestral animals, had an increased pneumonia score in the lungs following ancestral challenge on day 6 psc (Supplementary Fig. 7). Of note, the NL63 primary group had marked accumulation of red blood cells in the epithelium, blood vessels, and alveolar space on day 3 psc (Hemorrhage score of 2/4). Small clusters of mononuclear cells were evident in the parenchymal interstitium and, to a lesser extent, in the alveoli on day 3 and 6 psc (Fig. 3D and Supplementary Fig. 7). In general, the OC43 inoculated animals had lungs with less evidence of immune cell infiltration and pathology compared to NL63 inoculated animals, with lower inflammation scoring and proportion of the lung affected in OC43 inoculated animals on days 6 and 14 psc (Supplementary Fig 7), although epithelial sloughing in the bronchioles was evident on day 3 psc and mild interstitial pneumonia was present on day 6 psc in OC43 inoculated animals. Areas of pneumocyte hypertrophy were also observed lasting until day 14 psc (Fig. 3D). The Ancestral-Ancestral control group had little evidence of alveolar wall thickening and the alveolar space remained clear; however, there was significant presence of red blood cells in the alveolar wall and mononuclear cell infiltrates in the bronchiolar walls on day 3 psc. Some hyperplasia of alveolar pneumocytes were also noted on day 3 and 6 psc (Fig. 3D). No animal in the Ancestral-Ancestral group scored over a 2 on any lung histology severity marker, with an overall peak on day 3 psc at 1.3/4 (Supplementary Fig. 7). The damage to the lung as noted in the histopathology in the Mock-Ancestral group was accompanied by an increase in lung weight which peaked on day 6 psc at 5% of animal body weight (Fig. 3E). Ancestral-Ancestral animals had a significant decrease in lung weight compared to Mock-Ancestral animals on all days psc, whereas the seasonal coronavirus inoculated groups had significant decrease in lung weight compared to mock-inoculated animals on days 3 and 14 psc, but not day 6 psc (Fig. 3E).

While characterizing seasonal coronavirus infection in hamsters we found neutralizing antibodies toward OC43 or NL63 were elicited by day 14 pi. Therefore, we next wondered if neutralizing antibodies elicited after an NL63 or OC43 inoculation event would be increased or decreased during ancestral virus secondary challenge (Fig. 4A). In-life blood samples were collected from all animals in each group on day 14 ppi and again on day 55 ppi prior to ancestral virus challenge. After ancestral virus secondary challenge, blood samples were collected on days 59, 62, 65, and 70 ppi (3, 6, 9, and 14 psc). On day 14 ppi, the NL63 neutralizing antibodies from NL63 inoculated animals ranged in titer from 1:80 to 1:320 and OC43 inoculated animals had OC43 neutralizing antibody titers which ranged from 1:40 to 1:320. These initial titers decreased over the rest period prior to secondary challenge as shown by microneutralization assays utilizing plasma collected on day 55 ppi

(Fig. 4Ai, ii). Conversely, the ancestral virus inoculated animals had stable neutralizing antibody titers over the rest period (Fig. 4Aiii). Interestingly, during ancestral virus challenge, the NL63-Ancestral group had an increase or a back-boosting effect of NL63 neutralizing antibodies following ancestral virus exposure, with titers increasing above 1:320 by day 14 psc. The day 14 psc NL63 neutralizing antibody titers were significantly increased compared to day 14 ppi (1:160), as well as just prior to the secondary inoculation, on day 55 ppi, (1:80) (Fig. 4Ai). No back-boosting of OC43-specific antibodies was observed for the OC43 inoculated animals in the secondary challenge period as these titers remained stable (Fig. 4Aii). As expected, the Ancestral-Ancestral group had significant increases in ancestral-specific neutralizing antibodies during re-exposure (Fig. 4Aiii).

Additionally, since heterologous virus exposure-re-exposures events are known to elicit broadly reactive antibodies[43–45], we investigated the broadness of the elicited antibody response following secondary challenge, against the variants of SARS-CoV-2 (Fig. 4B and Supplementary Fig. 8). Microneutralization assays were performed using plasma isolated from all primary inoculation-challenge groups against ancestral SARS-CoV-2 as well as antigenically divergent variants, Alpha, Beta, Delta, and Omicron (BA.1). No statistical differences were detected in the neutralization of ancestral virus using the plasma collected from the Mock-Ancestral, NL63-Ancestral, or OC43-Ancestral groups, with all three groups reaching a maximum titer of 1:320 by day 14 psc (Fig. 4B). The homologous Ancestral-Ancestral group's plasma had the highest titers of cross-neutralizing antibodies on day 14 psc against all the variants, Alpha (1:1280) then Beta (between 1:2560 and 1:640) and Omicron (between 1:640 and 1:160). Cross-neutralizing antibodies against the VOCs were detected by day 14 psc in the Mock-Ancestral and seasonal coronavirus-Ancestral groups although at much lower titers across all variants examined compared to the Ancestral-Ancestral group (Fig. 4B). These results suggested that a heterologous inoculation-challenge exposure using the distant seasonal coronaviruses followed by ancestral SARS-CoV-2 did not induce substantial cross-protective antibodies as previously observed for heterologous influenza virus inoculation-challenge exposures[43–45]. However, both seasonal virus inoculated groups had non-significant trends of higher neutralizing antibody titers against the antigenically distant variants of Delta and Beta, compared to the mock-inoculated group (Fig. 4B and Supplementary Fig. 8).

### Previous infection with ancestral SARS-CoV-2 induces cross-protection against secondary challenge with antigenically divergent variants

Antigenically distinct SARS-CoV-2 variants circulate in people and can escape pre-existing immunity from ancestral virus infection and COVID-19 vaccines[21]; however, from sequence analyses (Fig. 1) the diversity within the SARS-CoV-2 virus lineages is recognized to be lower than that observed among the seasonal coronaviruses and ancestral SARS-CoV-2. Therefore, we leveraged the closer distances of the SARS-CoV-2 variants for primary inoculation-challenge studies to compare against the seasonal coronavirus-Ancestral primary inoculation-challenge combinations which had greater antigenic distances. We followed a similar experimental plan as used for the seasonal coronavirus-Ancestral experiments. A total of 18 male Syrian hamsters were used in each group (three animals sampled on days 0, 3,

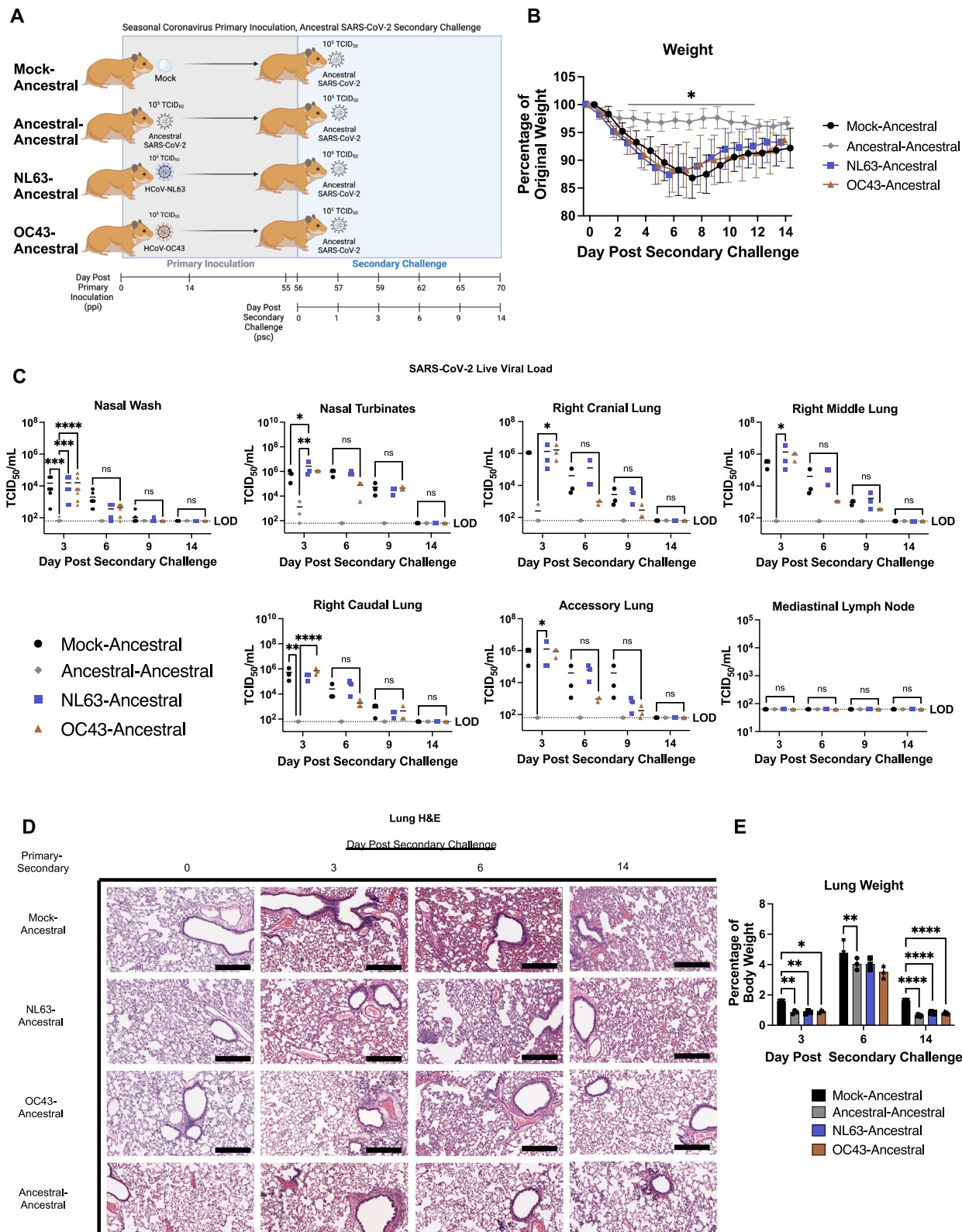

6, and 9 post secondary challenge and six animals sampled on day 14 post secondary challenge). Hamsters were intranasally inoculated with ancestral SARS-CoV-2 and then challenged on day 56 ppi with either Beta or Omicron (BA.1) VOCs (Fig. 5A), designated Ancestral-Beta or Ancestral-Omicron. Beta and Omicron were chosen as challenge viruses since they were the most antigenically divergent from ancestral SARS-CoV-2 at the time of the experimental design as determined by

cartography and serological studies[26]. These variants, which were antigenically distinct yet represented a small antigenic distance, would allow for complex immunological comparisons and mechanistic investigation against our previous seasonal-Ancestral inoculation-challenges. All inoculations were performed intranasally at an infectious dose of $10^5$ TCID$_{50}$. Heterologous secondary challenges were again compared to the control homologous Ancestral-Ancestral

**Fig. 3 | Inoculating with NL63 or OC43 offer little protection during ancestral SARS-CoV-2 challenge.** The schematic shows the study design for investigating the effects of inoculating with seasonal coronaviruses prior to a secondary SARS-CoV-2 ancestral virus challenge. Created with BioRender – Publication Licence IC25QNL6KM (**A**). Syrian hamsters were inoculated with a seasonal coronavirus NL63 or OC43 or mock inoculated with DMEM through intranasal instillation. As a comparison, one group of hamsters were inoculated with SARS-CoV-2 ancestral virus for a homologous infection-reinfection control. Nasal washes were collected throughout the acute phase to confirm infection. All animals were secondary challenged on day 56 ppi with ancestral virus through the intranasal route at $10^5$ TCID$_{50}$. Blood was collected on day 14 and 55 ppi (post primary inoculation) and on day 3, 6, 9, and 14 psc (post secondary challenge). Nasal washes, tissues and blood were collected to assess viral load, antibody dynamics, and pathology. Animal daily weight was assessed post secondary challenge. $n = 15$ animals per group (**B**). Live virus was assessed in nasal washes as well as throughout the respiratory tract by TCID$_{50}$ assay. $n = 15$ animals per group (**C**). Lungs were investigated for histopathological changes and fluid intake/immune cell infiltration by H&E histopathology (**D**) and assessment of lung to body weight ratio, respectively. $n = 15$ animals per group (**E**). At least 3 animals per timepoint were analyzed. ns indicates a $p$-value > 0.05, * indicates a $p$-value between 0.05 and 0.005, ** indicates a $p$-value between 0.005 and 0.0005, *** indicates a $p$-value between 0.0005 and 0.0001, and **** indicates a $p$-value < 0.0001 determined by one-way ANOVA comparing all groups on the same day. Data are presented as mean values with error bars indicating ±SD as appropriate. Slides were visualized and imaged using the Aperio ScanScope XT. Images were captured at 10X. Scale bars represent 300 μm. Images shown are representative of 3 animals per group, per timepoint. Source data are provided as a Source Data file.

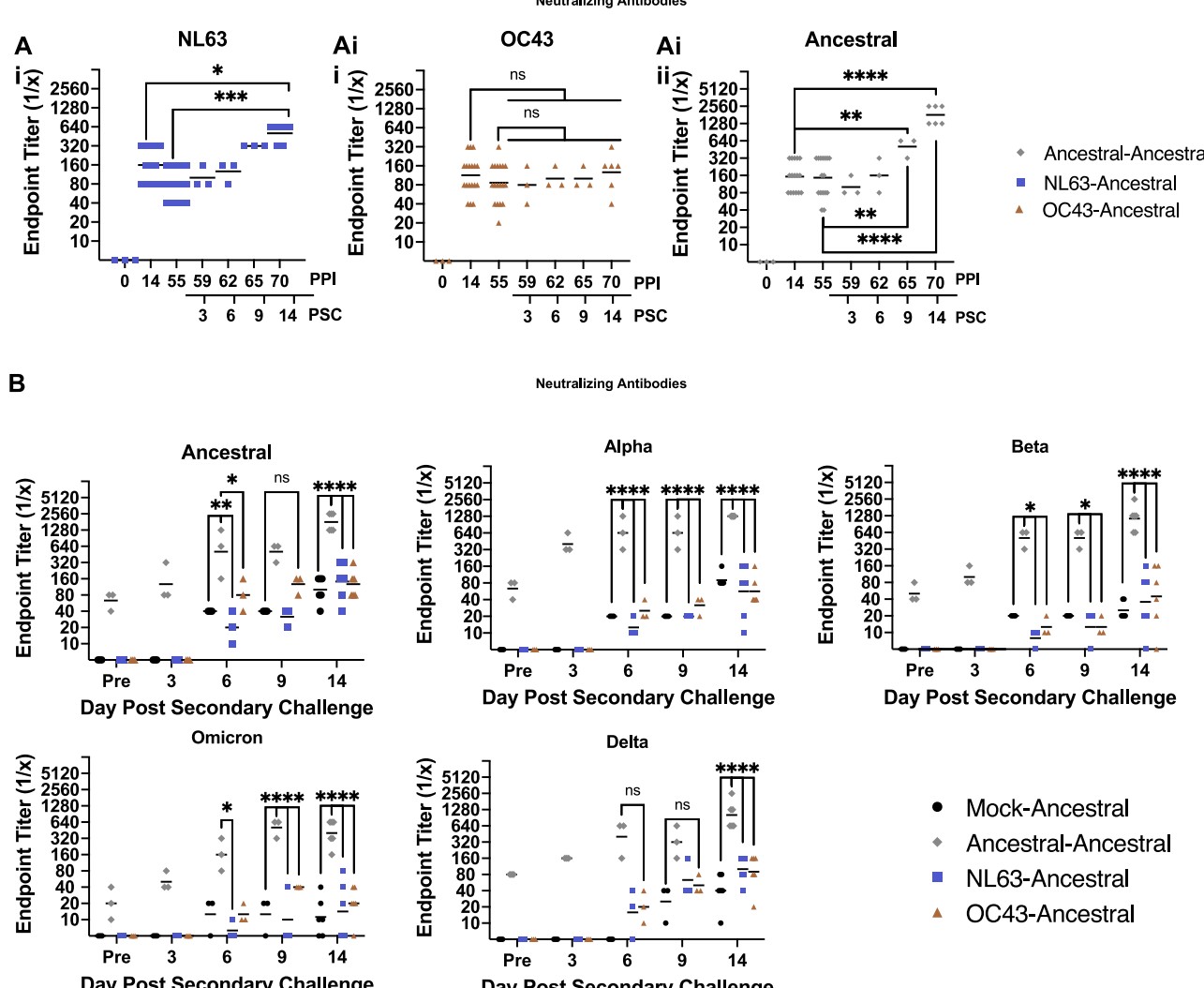

**Fig. 4 | Neutralizing antibodies elicited against seasonal coronaviruses decrease over time and show little cross-reactivity against SARS-CoV-2 variants.** Blood was collected from all inoculated animals on day 14 and 55 ppi and then from animals removed at specific time points following secondary ancestral virus challenge. Microneutralization assays were performed using the plasma samples collected from each group against their respective inoculating virus: NL63 (**Ai**), OC43 (**Aii**), and SARS-CoV-2 ancestral (**Aiii**). $n = 18$ animals per group sampled over 7 timepoints. Following SARS-CoV-2 ancestral virus secondary challenge, virus neutralizing antibodies were also quantified against SARS-CoV-2 ancestral virus and the variants Alpha, Beta, and Omicron to determine the potential for cross-neutralization. 'Pre' samples were taken prior to secondary inoculation and samples were taken on day 3, 6, 9, and 14 post secondary inoculation for assessment (**B**). $n = 18$ animals per group samples over 5 timepoints. ns indicates a $p$-value > 0.05, * indicates a $p$-value between 0.05 and 0.005, ** indicates a $p$-value between 0.005 and 0.0005, *** indicates a $p$-value between 0.0005 and 0.0001, and **** indicates a $p$-value < 0.0001 determined by one-way ANOVA comparing indicated days for **A** and comparing groups on the same day (**B**). Line represents group geometric mean. Source data are provided as a Source Data file.

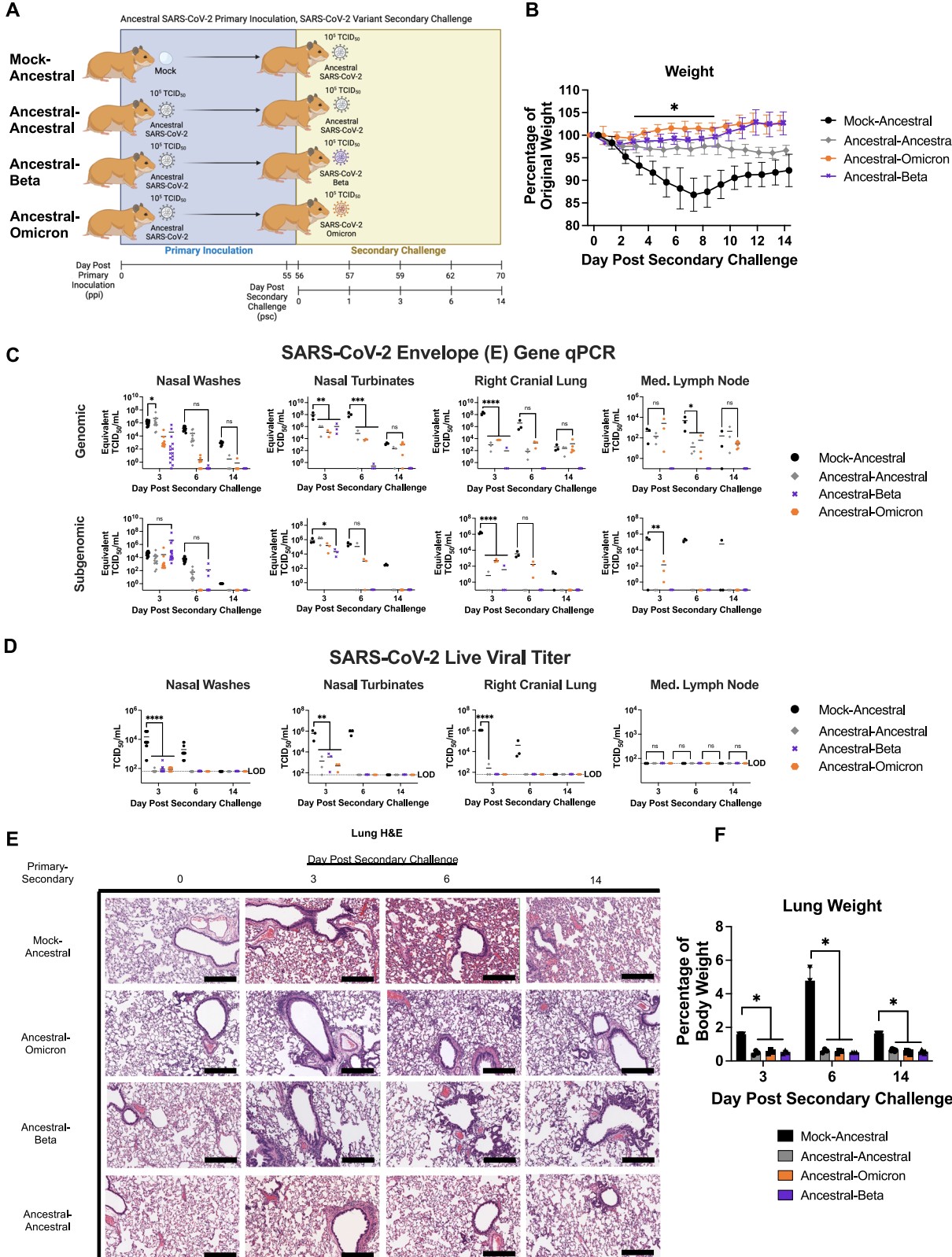

group. Following secondary challenge, both Beta and Omicron challenge groups had slight initial dip in weight (Fig. 5B). All groups had low levels of viral RNA post secondary challenge in the nasal washes as well as in the respiratory tissues (nasal turbinates and right cranial lung) and mediastinal lymph node (Fig. 5C). Subgenomic viral RNA was also detected until day 6 psc for some groups in the nasal washes, nasal turbinates, and lungs (Fig. 5C). However, live virus was only detected in

the nasal washes and nasal turbinates between $10^2$ and $10^3$ $TCID_{50}/mL$ on day 3 psc (Fig. 5D). One animal had detectable virus titers slightly above the limit of detection ($10^{1.8}$ $TCID_{50}/mL$) in the right cranial lung in the Ancestral-Ancestral group (Fig. 5D). Lung histopathology indicated peribronchiolar mononuclear cell infiltration on day 3 psc of all groups, while the alveolar space of the Ancestral-Omicron, Ancestral-Beta, and control Ancestral-Ancestral groups remained mostly clear

**Fig. 5 | SARS-CoV-2 ancestral virus inoculating provides significant protection during a secondary heterologous challenge with antigenically distinct SARS-CoV-2 variants Beta or Omicron.** The primary infection and secondary challenge study plan illustrates the experimental groups and time points assessed. Created with BioRender – Publication Licence IC25QNL6KM (**A**). Syrian hamsters were inoculated with SARS-CoV-2 ancestral virus at $10^5$ TCID$_{50}$ and received a secondary challenge on day 56 with $10^5$ TCID$_{50}$ Omicron, Beta, or homologous ancestral SARS-CoV-2. Samples were collected for viral load and pathology. Daily weights were recorded for 14 days post secondary inoculation. $n = 15$ animals per group (**B**). Viral RNA by qRT-PCR was assessed post secondary challenge in nasal washes, respiratory tract tissues, and mediastinal lymph node $n = 15$ animals per group (**C**). TCID$_{50}$ assays were used to assess live virus in nasal washes as well as throughout the respiratory tract $n = 15$ animals per group (**D**). Histopathology was conducted to determine pathological changes prior to secondary challenge, and on day 3, 6, and 14 psc (**E**). Lung to body weight ratio was also investigated $n = 15$ animals per group (**F**). At least 3 animals per timepoint were analyzed. ns indicates a $p$-value > 0.05, * indicates a $p$-value between 0.05 and 0.005, ** indicates a $p$-value between 0.005 and 0.0005, *** indicates a $p$-value between 0.0005 and 0.0001, and **** indicates a $p$-value < 0.0001 determined by one-way ANOVA comparing groups on the same day. Data are presented as mean values with error bars indicating ±SD as appropriate. Slides were visualized and imaged using the Aperio ScanScope XT. Images were captured at 10X. Scale bars represent 300 μm. Images shown are representative of 3 animals per group, per timepoint. Source data are provided as a Source Data file.

with little evidence of leukocyte infiltration and hemorrhage in the area (Hemorrhage and Bronchus severity scores of 0 throughout the time course. (Fig. 5E and Supplementary Fig. 9). Minimal epithelial sloughing was also noted in the Ancestral-Omicron and Ancestral-Beta groups but was not detected in the Ancestral-Ancestral group (Fig. 5E). Both Ancestral-variant groups had an increase in overall score compared to Ancestral-Ancestral at select timepoints throughout secondary challenge. For the Ancestral-Omicron group specifically, one animal scored a 4/4 on proportion, meaning over 75% of the lung was affected, while no more than 50% and 25% of the lung was affected in the Ancestral-Beta and Ancestral-Ancestral groups, respectively (Supplementary Fig. 9). Lung weights from all groups at necropsy days post secondary challenge were significantly below that of the Mock-Ancestral group (Fig. 5F).

We next investigated the cross-reactivity and cross-neutralizing ability of the plasma antibodies elicited following secondary challenge against the viruses in the SARS-CoV-2 lineage: ancestral strain, Alpha, Beta, Delta, and Omicron (Fig. 6A). For all evaluations, the Ancestral-Omicron group trended toward having the lowest neutralizing titers against the panel of SARS-CoV-2 variants except against the Omicron virus itself. The Ancestral-Omicron group had neutralizing titers on day 14 psc with geometric means >1:320, >1:320, >1:320, and >1:640, for the ancestral strain, and Beta, Alpha, and Delta variants, respectively (Fig. 6B). The Ancestral-Omicron group had titers of 1:1280 against the Omicron virus itself. Ancestral-Ancestral and Ancestral-Beta groups were above 1:640 for all variants with the exception of the Omicron neutralization assay which were lower (Fig. 6B). Interestingly, the Ancestral-Beta group had a higher median than the Ancestral-Ancestral group for the neutralization of Beta, Omicron, Alpha, and Delta viruses (Fig. 6B). These results suggest cross-protective responses where Beta and ancestral SARS-CoV-2 are antigenically closer to each other than to Omicron. Furthermore, heterologous exposure of ancestral virus followed by Beta may be an optimal antigenic distance between the two viruses to induce a greater breadth of antibody cross-reactivity.

## Host response analysis during secondary challenge indicates immune regulation is based on antigenic distance

Given the various clinical outcomes of the primary inoculation-secondary challenge combinations, we went on to investigate the potential immunological mechanisms driving pathology, viral clearance, and antibody specificity using host gene expression profiling by quantitative real-time PCR (qRT-PCR) in the nasal turbinates (Fig. 7A and Supplementary Fig. 10), right cranial lung (Fig. 7B and Supplementary Fig. 11), and mediastinal lymph nodes (Fig. 7C, Supplementary Fig. 12, and Supplementary Fig. 13). Our established qPCR panels were designed to examine specific compartments of the immune system including antiviral responses, inflammatory responses, and adaptive T and B cell immunity[35]. For the antiviral responses, type I interferon-related (*IFN-β*, *STAT2*, *IRF1*, *IRF3*, and *TLR3*) and type II interferon-related (*IRF2*, *STAT1*, *CXCL10*, and *IFN-γ*) responses were evaluated. *IL-6*, *IL-1β*, and *TNF* were examined regarding inflammation. T cell responses were examined per specific T cell subset profile which included general T cell-related genes (*CD3*, *CD4*, *CD8*, *IL-2*); follicular T helper cell (*IL-21* and *CXCR5*); cytotoxic T cell markers and cytotoxic mediators (*CD8*, *PRF1*, and *GZMB*); T$_{H1}$ responses (*IFN-γ*, *IL-12*, and *T-bet*); T$_{H2}$ responses (*GATA-3*, *IL-4*, *IL-5*, and *IL-13*); T$_{H17}$ responses (*ROR-γ-T*, *IL-17*, and *IL-22*); and T regulatory (T$_{REG}$) responses (*FOXP3*, *IL-10*, and *TGF-β*)[35]. B cell responses were also examined through the regulation of B cell-related markers and cytokines (*CD19*, *AID*, *BCL6*, and *IL-6*). For all analyses, fold change is displayed in heat maps or bar graphs (Fig. 7 and Supplementary Figs. 10–13). To determine profile trends along antigenic distance, the profiles were organized dependent on the antigenic relationship between the primary inoculation and challenge virus with the most distant combinations placed on the left and the most similar (the homologous inoculation-challenge) on the right (Fig. 7).

Analysis of the antiviral response indicated that the Mock-Ancestral (negative control) group had downregulation of type I interferon response genes in the nasal turbinates post secondary challenge (Fig. 7A). Similarly in the lung, on day 3 psc, *IFN-β* had a decreased fold change of 0.03 and *IRF1* had a fold change of 0.6 (Fig. 7B). Interestingly, the seasonal coronavirus inoculated groups also had significant suppression or downregulation of type I interferon gene expression after ancestral virus challenge, illustrated by similar decreases of *IFN-β*, *STAT2*, *IRF1*, *IRF3*, or *TLR3* in the upper and lower respiratory tract. Additionally, there were no significant differences in the regulation of type II/general interferon response genes *IFN-γ*, *IRF2*, *STAT1*, and *CXCL10* between the mock inoculated animals and the NL63 primary or OC43 inoculated animals at challenge (Fig. 7A, B).

In contrast to the profile observed in the seasonal coronavirus inoculated groups, animals inoculated with ancestral SARS-CoV-2 and subsequently challenged with a variant had greater type I interferon responses throughout the respiratory tract (Fig. 7A, B). In the nasal turbinates, upregulation of *IFN-β* was observed for the groups challenged with Omicron, Beta, and ancestral virus at fold changes of 3.1, 5.2, and 5.7, respectively (Fig. 7A). In the lungs, *IFN-β* was upregulated 3.8-fold (day 3 psc) for the Ancestral-Ancestral group, 3.9-fold for the Ancestral-Beta (day 6 psc), and 2.1-fold (day 3 psc) for the Ancestral-Omicron group (Fig. 7B). In respect to *IFN-γ*, the Omicron challenge group had a fold increase of 6.1 (day 6 psc) in the lungs which was significantly less of an increase than the 16.3-fold increase in the control Mock-Ancestral group (Fig. 7B). Minimal increases in IFN-γ in the ancestral primary inoculated groups following Beta or ancestral secondary challenge were found (Fig. 7B). Together this analysis suggests the antiviral response to be positively regulated during secondary challenge when the primary inoculation and challenge virus are antigenically similar and the host antiviral responses to be inhibited when the sequential coronavirus exposures are antigenically distant.

We next investigated T helper cell responses by examining the regulation of T$_{H1}$, T$_{H2}$, T$_{H17}$, and T$_{REG}$ cell-related genes in the respiratory tract (Fig. 7A, B). Within the nasal turbinates, the T$_{H1}$ response was present across all groups, evidenced by upregulation of *T-bet*, the T$_{H1}$ master transcriptional regulator[46], as well as *IL-12*. In contrast,

**Fig. 6 | Heterologous SARS-CoV-2 variant inoculating and challenge increases the potential for cross-neutralization by elicited antibodies.** Hamsters were inoculated with SARS-CoV-2 ancestral virus then challenged with Beta, Omicron, or ancestral SARS-CoV-2 as control. Neutralizing antibodies in plasma samples were evaluated against SARS-CoV-2 variants following secondary challenge. ns indicates a $p$-value > 0.05, * indicates a $p$-value between 0.05 and 0.005, ** indicates a $p$-value between 0.005 and 0.0005, *** indicates a $p$-value between 0.0005 and 0.0001, and **** indicates a $p$-value < 0.0001 determined by one-way ANOVA comparing all groups to the Mock-Ancestral group. $n = 15$ animals per group sampled over 4 timepoints (**A**). Day 14 psc values were statistically compared for Ancestral-Ancestral, Ancestral-Beta, and Ancestral-Omicron groups. $n = 6$ animals per group sampled on day 14 psc (**B**). Line represents group geometric mean. Source data are provided as a Source Data file.

$T_{H2}$-related genes were upregulated in the Ancestral-variant groups, but not in the Mock-Ancestral or seasonal-Ancestral groups. In particular, significant increases in *IL-5* and *IL-13* were noted in the Ancestral-variant groups, and not in other combinations, in the nasal turbinates (Fig. 7A). Upregulation of $T_{H17}$ associated genes was noted in the nasal turbinates of the Mock-Ancestral and more antigenically distant seasonal coronavirus-Ancestral groups compared to the antigenically similar Ancestral-variant groups. *ROR-γ-T* was upregulated on day 3 psc in all groups regardless of challenge; however, only the seasonal coronavirus-Ancestral groups and the negative control Mock-Ancestral group had significant upregulation of $T_{H17}$ cytokines *IL-17* and *IL-22*. Fold changes between 15 and 19 were noted for *IL-17* and between 14 and 20 for *IL-22* for these groups.

Due to the association of uncontrolled inflammation with severe COVID-19 and unfavorable outcomes, we investigated *IL-6*, *IL-1β*, and *TNF* as inflammatory markers in the respiratory track[47–54]. Mock-Ancestral animals had peak increases of *IL-1β* (fold change of 26) and *TNF* (fold change of 9) on day 3 psc in the nasal turbinates (Fig. 7A).

NL63 and OC43 inoculated animals also had increases in these inflammatory markers in the time course; however, TNF was down-regulated for the OC43 inoculated group on day 3 psc. *IL-6*, which is known to be upregulated during severe COVID-19[55], was consistently upregulated with peak fold changes of 61.2, 57.3, and 60.1 for mock, NL63, and OC43 inoculated groups, respectively. Significant increases in *IL-6* were not observed in the Ancestral-Omicron, Ancestral-Beta, and Ancestral-Ancestral groups. Increases in these inflammatory markers for the mock and seasonal inoculated groups were also noted in the lungs (Fig. 7B). The increased inflammation in the seasonal coronavirus-Ancestral groups at secondary challenge suggests lack of protection coinciding with pathogenesis in the secondary challenge period.

Following viral infection, antigen is taken to the local draining lymph nodes to stimulate B cells for antibody production and germinal center reactions[56]. These responses occur faster during a secondary exposure if the primary and secondary antigens are similar. In an effort to understand the stimulation of cross-reactive antibodies, or lack

# Host Gene Expression qPCR

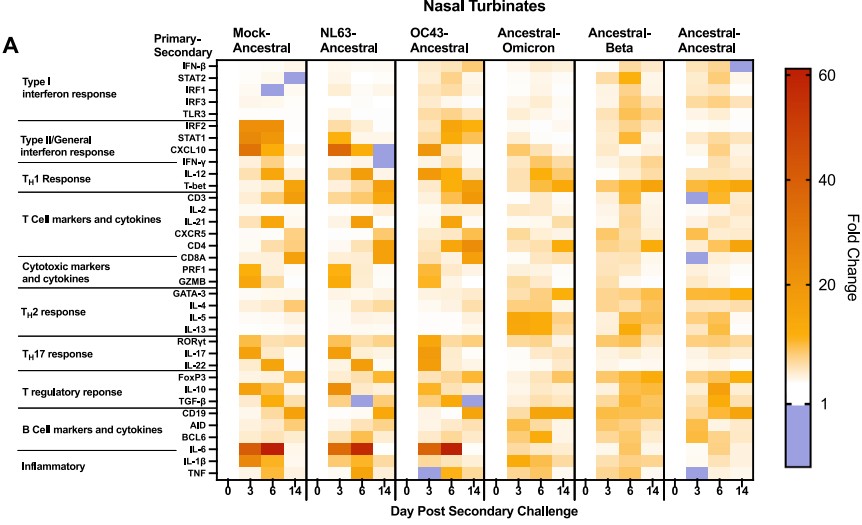

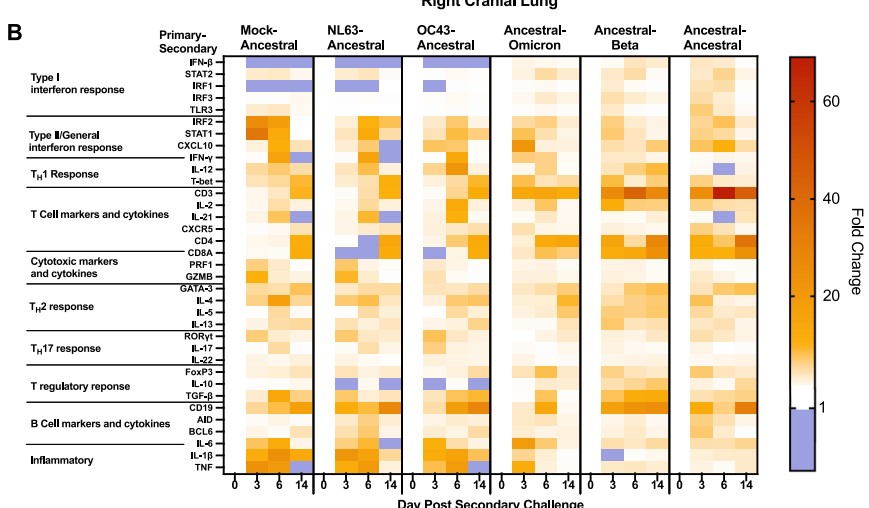

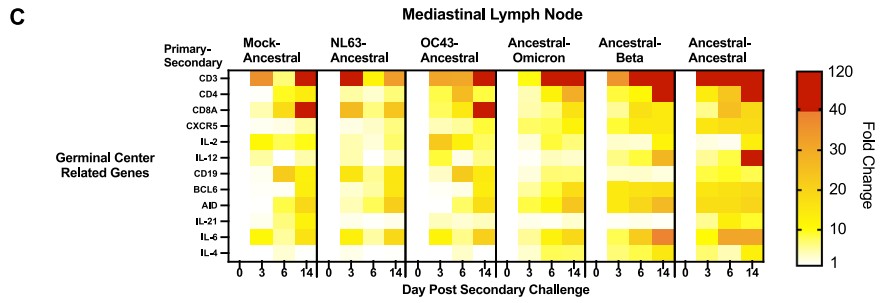

**Fig. 7 | Type I interferon, inflammation, and adaptive immune responses in the respiratory tissues upon secondary coronavirus challenge are regulated of according to antigenic distance of the inoculating and challenge virus.** qRT-PCR was performed on RNA extracted from nasal turbinate (**A**), right cranial lung (**B**), and mediastinal lymph node (**C**) tissue following secondary challenge in coronavirus inoculated hamsters. Gene regulation was analyzed according to response type and results were displayed as a heat map according to fold regulation. Groups are arranged according to antigenic distance between the inoculating and secondary challenge virus. Genes were organized by type I interferon response (*IFN-β, STAT2, IRF1, IRF3,* and *TLR3*), type II/general interferon response (*IRF2, STAT1, CXCL10,* and *IFN-γ*), TH1 response (*IFN-γ, IL-12,* and *T-bet*), T cell makers and cytokines (*CD3, IL-2, IL-21, CXCR5, CD4,* and *CD8A*), cytotoxic markers and cytokines (*CD8A, PRF1* and *GZMB*), TH2 response (*GATA-3, IL-4, IL-5,* and *IL-13*), T regulatory response (*FoxP3, IL-10,* and

*TGF-β*), B cell markers and cytokines (*CD19, AID, BCL6,* and *IL-6*), and inflammatory cytokines (*IL-6, IL-1β,* and *TNF*). Primers were designed or acquired specific to hamster genes (Table 2). Fold-change was calculated via ΔΔCt against baseline (Day 0) with *BACT* as the housekeeping gene. The legend depicts fold change in which upregulation (>1) is represented ranging in from white (=1) to red. Any down-regulation (<1) is represented in blue. At least three animals were analyzed for each timepoint. Statistical differences from Mock-Ancestral group can be found in Supplementary Table 4 for nasal turbinates, Supplementary Table 5 for right cranial lung and Supplementary Table 6 for the mediastinal lymph node. Bar graphs of this data can be found in Supplementary Fig. 10 for nasal turbinates and Supplementary Fig. 11 for right cranial lung. A full heatmap of the mediastinal lymph node data can be found in Supplementary Fig. 12 and the corresponding bar graphs are in Supplementary Fig. 13. Source data are provided as a Source Data file.

there-of, as well as the antigenic relationship between the primary seasonal coronaviruses and SARS-CoV-2, the mediastinal lymph nodes collected on day 3, 6, and 14 psc were investigated for key drivers of the germinal center reaction (Fig. 7C). Specifically, we investigated the regulation of *CD3*, *CD4*, *CD8*, *IL-2*, *IL-12*, *CD19*, *IL-21*, *IL-6*, and *IL-4* gene expression as we have done previously for primary SARS-CoV-2 infection in Syrian hamsters[35]. We also added the analysis of *CXCR5* which is expressed on T follicular helper cells, *BCL6* which is the master transcription regulatory factor of both T follicular helper (T$_{FH}$) cells and germinal center dependant B cells, and *AID* (activation-induced cytidine deaminase) which is necessary for the development of mature, class-switched antibody isotypes[57–59]. Similar to our published results in naïve hamsters[35], we found genes associated with adaptive immune responses, *CD3*, *IL-2*, *IL-12*, *CD19*, and *IL-21*, to have biphasic regulation in the lymph nodes during a primary challenge (Mock-Ancestral; Fig. 7C). In stark comparison, the homologous Ancestral-Ancestral group had early, strong upregulation of the majority of the genes, which increased in intensity over day 6 and 14 psc instead of exhibiting biphasic expression. When comparing these profiles to the NL63-Ancestral and OC43-Ancestral groups, the profiles of germinal center gene expression shared characteristics of both the homologous reinfection group and the naïve Mock-Ancestral group. NL63-Ancestral animals had early strong *CD3*, *CD8*, *IL-6* responses at secondary challenge but also showed biphasic expression of many genes including *CD4*, *CD8*, *IL-2*, *IL-12*, *CD19*, and *IL-21*. In contrast, the OC43-Ancestral group had strong *CD3* upregulation throughout the ancestral challenge time course which peaked on day 14 psc. *CD8* levels were also high in the OC43-Ancestral group on day 14 psc but more comparable to the mock-inoculated group. We analyzed the B cell (discontinuous) (Supplementary Fig. 14A) and T cell (linear) (Supplementary Fig. 14B) epitopes among the seasonal coronaviruses in our study in comparison to ancestral SARS-CoV-2 and VOCs to give insight into our virological and immunological results especially that of OC43. B-cell epitope predictions were done to identify surface epitopes. This analysis suggested that discontinuous B cell epitopes are less accessible on the S protein for the seasonal coronaviruses compared to ancestral SARS-CoV-2 and variants. For T cell MHC I and MHC II epitope analysis, the data is displayed as a percentage of similar epitopes for the S proteins of coronaviruses against the ancestral spike sequence and compared to the Glycoprotein GP120 for HIV-1 and Hemagglutinin (HA) for the influenza virus H7N9 as a negative control. Supporting our gene expression analysis, OC43 had more epitopes binned in the highest category compared to NL63. Although the T cell analysis was done against human MHC molecules and our experimental data was performed in hamsters, this analysis may suggest there to be more cross-reactive T cell responses from an OC43 previous infection with ancestral SARS-CoV-2 than from a previous NL63 infection.

The more antigenically similar groups of Ancestral-Omicron and Ancestral-Beta had profiles that were not regulated in a biphasic distribution. For example, *CXCR5* was upregulated on day 3 psc and remained high throughout the time course for these groups (Fig. 7C). These results suggest induction of sustained gene expression was associated with the recall of an adaptive response during a secondary exposure with an antigenically similar coronavirus while biphasic gene expression was associated with antigenically dissimilar viruses and the induction of a more de novo adaptive response.

## Discussion

Coronaviruses are a significant threat to public health, animal welfare, and economic stability as new strains emerge and endemic coronaviruses such as SARS-CoV-2 and seasonal coronavirus species continue to circulate in humans. The path to controlling the public health burden of coronaviruses involves understanding immune protection against circulating and emerging strains, as well as the cross-reactivity among these viruses[60]. In our study, we wanted to determine

how a previous infection with a coronavirus, or in other words how being imprinted with a seasonal coronavirus, impacted the clinical and immunological outcome of an antigenically similar and antigenically distant secondary coronavirus challenge. To this end we took a step-wise approach to determine the cross-reactivity and cross-protection induced by imprinting from a primary infection and then assessed protection and cross-reactivity at challenge. We performed studies in Syrian hamsters leveraging seasonal coronaviruses to represent antigenically distant viruses and SARS-CoV-2 variants of concern (VOCs) to represent antigenically similar viruses against the prototypic SARS-CoV-2 ancestral virus. We found imprinting with seasonal coronaviruses offered little protection against an ancestral virus challenge, yet the immune responses to challenge differed in the NL63 imprinted group compared to the OC43 imprinted animals. There was a back-boosting of NL63 virus-neutralizing antibodies in the NL63 imprinted animals during SARS-CoV-2 challenge. In contrast, OC43 imprinted animals exhibited trends toward faster viral clearance as well as gene expression and T cell epitope similarities suggestive of potentially cross-reactive T cell responses toward secondary ancestral virus challenge. Ancestral virus imprinting studies indicated secondary challenge with the Beta variant to induce the most favorable antibody responses in terms of broad reactivity. We also found that antibodies elicited to seasonal coronaviruses to have decreased durability suggesting immunity to seasonal coronaviruses is short-lived, as previously reported[9,11,29]. Since only human serological studies on the imprinting effects of seasonal coronaviruses have previously been performed, our study expands on these findings by isolating the immunological impacts of specific coronaviruses[11,15,28–30,61–67]. Our results bring valuable insight into the specific contributions of pre-existing immunity from coronavirus infection and the associated mechanisms of regulation.

Pre-existing humoral immunity gained by prior infection or vaccination dictates protection from reinfection and disease development[68]. Understanding the mechanisms of imprinting and immune recall is essential for developing vaccines and public health strategies. Previous studies in humans focusing on the impact of pre-existing humoral immunity to coronaviruses found SARS-CoV-2 nucleocapsid (N) and S antibodies to be present in as many as 20% of SARS-CoV-2 unexposed study participants[15,28]. This suggested that a possible previous exposure to other viruses such as the seasonal coronaviruses 229E, NL63, HKU1, or OC43 was responsible for the presence of these SARS-CoV-2 cross-reactive antibodies prior to the COVID-19 pandemic. Additional analysis found the majority of the pre-existing antibodies to specifically react to OC43 and HKU1 while NL63 antibodies were least represented[15,28,69,70]. Further investigation found SARS-CoV-2 infection to back-boost pre-existing OC43 and HKU1 S2 domain antibodies during the secondary exposure suggesting cross-reactivity between OC43 and HKU1 with SARS-CoV-2 in this membrane proximal region of the S protein[15,28,69,70]. Our study of experimentally imprinted and challenged hamsters found that NL63 neutralizing antibodies were back-boosted during ancestral virus challenge; however, there was concomitant higher viral load in NL63 imprinted animals compared to OC43 imprinted animals, although not statistically significant. Since our study only focused on neutralizing antibodies and not on non-neutralizing antibodies or epitopes as done in the previous human studies, this may explain why we did not detect OC43 antibody back-boosting. This was surprising since OC43 and SARS-CoV-2 are both betacoronaviruses. It is possible that back-boosted neutralizing antibodies were observed in the NL63 imprinted animals because NL63 and SARS-CoV-2 both use ACE2 and blocking host cell receptor binding may rely on similar virus neutralizing epitope involved in ACE2 docking. We also found high viral load during SARS-CoV-2 challenge in the NL63 imprinted animals suggesting an inefficient immune recall response which may represent Original Antigenic Sin. Findings from Wratil and colleagues also suggested Original

Antigenic Sin to occur in people with pre-existing immunity to alpha-coronaviruses during SARS-CoV-2 infection leading to a greater incidence of COVID-19[29].

Our study design had a relatively short recovery time of 56 days between primary infection and secondary challenge as it was long enough for allow innate responses to return to baseline and adaptive responses to begin contracting while also being short enough for data acquisition and competition of our many groups. However, our conclusion can only reflect responses in this timeframe. Since the acute infection phase of COVID-19 in humans is considered to be 2–4 weeks[71,72], to understand the development of long-lived memory cells and the recall response upon secondary exposure, a longer recovery time should be studied. In our study, we observed a decrease in antibodies against NL63 and OC43 over the recovery period, which was not observed for the ancestral SARS-CoV-2 infected group. It would be interesting to investigate longer time periods for the seasonal coronaviruses as we expect that antibody levels would continue to decline, as has been seen in humans[11,73,74]. Given this, upon exposure to SARS-CoV-2 at a later timepoint when antibody responses are waning, it is possible that less of an antigenic sin-like response towards NL63 is induced. Several studies have now investigated long-term memory development against SARS-CoV-2 in humans[75–79]. Dan et al.[75] demonstrated that 8-months after infection, B cell responses were relatively maintained in patient sera, while the CD4 and CD8 T cell responses decreased over time suggesting that if we extended our recovery period after SARS-CoV-2 exposure the humoral response may be sustained, while the decrease of CD4 cells over time may affect the germinal center response through a T follicular helper cell-dependent mechanism[80–82]. More work is needed to understand how cross-reactivity among coronaviruses may change over time.

Although we did not find evidence of cross-neutralizing antibodies in OC43 inoculated animals, our data suggested possible recall of cross-protective T cell responses during SARS-CoV-2 challenge since the virus in the respiratory tract appeared to clear somewhat faster than the NL63 or mock-inoculated groups. Previous studies have suggested that infection with seasonal betacoronaviruses primes the human immune system for protective yet non-neutralizing responses against SARS-CoV-2[30]. Specifically, we found that OC43 imprinting was associated with stronger or earlier increases in T cell-related genes such as *CD3*, *IL-2*, *IL-12*, *IL-21*, *CXCR5*, *CD4*, and *CD8A* in the lungs and *CD3* and *CD4* responses in the mediastinal lymph node. These profiles were more similar to the T cell profiles from the antigenically closer Ancestral-variant inoculation-challenge groups. We also identified by immunoinformatics a higher percent similarity of MHC class I and MHC class II T cell epitopes between OC43 and the ancestral virus compared to NL63 and ancestral virus, supporting our T cell response signatures. Pre-existing cross-reactive CD8+ and CD4 + T cells have been shown to be induced during SARS-CoV-2 infection in people[11,62–66] and as many as 40–60% of unexposed individuals have cross-reactive T cell responses even before SARS-CoV-2 exposure[63–65]. Other studies have provided evidence that cross-reactivity stems from previous exposure to 229E and OC43[62,64,65]. Although much has been done in humans to understand pre-existing T cell immunity, insufficient effort to date has been made to determine the specific relationship between the seasonal coronaviruses of NL63, OC43, 229E, and HKU1 and regulation of cross-reactivity with SARS-CoV-2. The present study did not include analysis of HKU1. Previous exposure to HKU1 will require additional follow up which would provide interesting insight. HKU1 is also a betacoronavirus and is second only to OC43 for the seasonal coronaviruses in terms of spike amino acid identity when compared to ancestral SARS-CoV-2 (Fig. 1A). HKU1 uses the same entry receptor as OC43, 9-O-acetylated sialic acid. With these commonalities, in addition to having 63.3% identity to OC43 spike (Fig. 1A), it is possible that HKU1 inoculation could have a similar effect on secondary exposure to lineage B SARS-CoV-2, through influence of the T cell response.

Follow-up studies leveraging the hamster model of seasonal coronavirus infection we have put forth here could be used to resolve the unique effects of specific seasonal coronaviruses with SARS-CoV-2 cross-reactive T-cell responses.

Coronaviruses have a robust ability to inhibit the host antiviral responses and interferon signaling but how this impacts the establishment of the host's adaptive immune responses and recall responses during a secondary homologous or heterologous challenge has not been fully elucidated[83]. Interestingly, the NL63-Ancestral, OC43-Ancestral, and Mock-Ancestral groups in our study all had inhibition of the type I interferon responses (*IFN-β* and *IRF1*) during ancestral SARS-CoV-2 secondary challenge, similar to our previous results with naïve SARS-CoV-2 infected hamsters[35]. This suggested that previous seasonal coronavirus infection was not able to inhibit the secondary challenge ancestral virus from blocking the antiviral response. Furthermore, the inflammatory genes *IL-6*, *IL-1β*, and *TNF* were more pronounced in the seasonal coronavirus imprinted animals during the ancestral secondary challenge with the NL63 group having the highest levels. The presence of inflammation also supported a lack of cross-protection from seasonal coronavirus imprinting, especially NL63. When looking at the antiviral responses in the Ancestral-variant groups, these signatures were not inhibited but instead were upregulated suggesting the ability of the previously exposed host to produce effective antiviral responses to viral infection. However, while imprinting with ancestral and challenging with the variants Beta or Omicron led to a significant amount of protection from infection and disease, the challenges did not induce the identical adaptive immune signatures as our homologous Ancestral-Ancestral group. The variations in adaptive gene signatures at secondary challenge indicated different immune recall dynamics. Since low levels of live virus and some evidence of lung damage were detected in these groups, we can conclude that the recall response or the present circulating immunity did not provide sterilizing quantities of neutralizing antibodies during the secondary exposure. Focusing on the antiviral response as a measure of protection from challenge acquired from imprinting, the Beta challenged hamsters had upregulation of the antiviral responses more similar to the Ancestral-Ancestral group. Conversely, there was minimal upregulation of type I interferon genes in the omicron-challenged animals who also had increases in *IL-6*, *IL-1β*, and *TNF*, similar to the inflammatory profiles of the mock control group. These results for the Omicron challenge group supports published findings that suggest Omicron to be more antigenically distant from ancestral than Beta or Alpha variants[21–27].

With human clinical data examining multiple exposures to coronaviruses[9,63,64,84,85], including SARS-CoV-2[86–91], being inconsistent due to host factors and co-morbidities, it is essential to have the ability to study infection-reinfection in appropriate animal models for experimental variable isolation. Ferrets[92,93], hamsters[36,74,94–99], and non-human primates[100,101], have all been used to study secondary challenge with SARS-CoV-2. While a primary exposure to the SARS-CoV-2 ancestral strain in rhesus macaques can lead to weight loss, viral load in respiratory and gastrointestinal tracts as well as lung damage, reinfection with the same virus on day 28 post primary inoculation led to protection[100]. Weight was not affected, and no live virus was recovered from any tissue collected, similar to our findings[100]. In the same non-human primate model, reinfection was assessed with SARS-CoV-2 variants Alpha and Beta and revealed that while a previous exposure to the ancestral strain did improve protection compared to naïve animals, viral load was still detected upon heterologous reinfection[101]. In homologous reinfection experiments with ferrets, secondary challenge with SARS-CoV-2 ancestral virus on day 28 or 56 post initial inoculation still maintained protection with no clinical signs reported or viral load detected[92,93]. Multiple studies have assessed infection-reinfection with the ancestral SARS-CoV-2 as well as variants revealing that while even heterologous reinfection with different

variants decreases clinical signs and the presence of live virus, previous exposure to a SARS-CoV-2 virus does not lead to complete protection[36,74,94–99]. Hansen et al, assessed homologous reinfection with SARS-CoV-2 ancestral virus, with the secondary challenge occurring on day 14, 49, and 152 days post primary and found that day 49 reinfection, the time point closest to our experimental design with a day 56 reinfection, was the most protective[97]. Shiwa-Sudo and colleagues observed increases in T cell activity correlated with earlier viral clearance[98]. These findings were consistent not only with our variant reinfections where we observed increases in T cell-related gene expression, but also in OC43-Ancestral animals where virus levels were lower in the nasal turbinates and lung compared to Mock-Ancestral animals, although noted not statistically significant. Assessment of host gene immune regulation has most often been in the case of single infections[102–105]. O'Donnell and colleagues assessed host response to primary infection with the SARS-CoV-2 viruses, the ancestral strain, Alpha variant, and Beta variant. They found that inflammatory signatures varied with between variants, with Alpha infection eliciting the most robust upregulation in inflammatory genes[103]. In our analysis, we also found differences in inflammatory signatures in response to the secondary inoculation with the variants with Omicron having an upregulation of *IL-6* and *TNF*, a finding we attributed to a greater antigenic distance between the imprinting and challenge viruses. It is important to note that the variants do cause different immune signatures even in a single exposure. Additionally, we found a high level of cross-neutralizing antibodies among SARS-CoV-2 ancestral virus and the VOCs. Previous studies have also investigated cross-protection and cross-neutralization among SARS-CoV-2 VOCs, with some studies showing high levels of cross-neutralization and others with lower levels[95,98]. Differences in cross-neutralization or protection compared to the findings from our study could be due to lower infectious doses at inoculation or time between primary infection and secondary challenge or blood collection. Future infection-reinfection studies should expand upon this work to understand how antigenically distant coronaviruses can impact the protection of one another.

Our present study established Syrian hamster preclinical models of seasonal coronavirus infection as well as imprinting. Additionally, we provided a direct comparison of the host responses during primary infection as well as secondary challenge to a full spectrum of seasonal and SARS-CoV-2 variant viruses informing immune mechanisms driving pathogenicity and protection. However, we were not able to access infection and host responses of the betacoronavirus HKU1 due to lack of availability of the virus. We also recognize that due to the lack of immunological reagents for Syrian hamsters, our immune response analysis was limited to host gene expression. Future work should aim to investigate immune cell subset dynamics during seasonal coronavirus infection and SARS-CoV-2 challenge either in the hamster or mouse model to complement our host response analysis.

Immunological imprinting, previously described as 'Original Antigenic Sin' or OAS, is the powerful mark, positive or negative, that a first viral infection makes on the host's immune system and the resulting response to subsequent antigenically related but distinct viral exposures[68,106,107]. Although we recognize the contributions of pre-existing immunity and its contribution to an adaptive immune response, the specific mechanisms regulating imprinting and recall during sequential exposures are not understood. Additionally, the potential of OAS to occur for coronaviruses has not been extensively studied as most research as focused on influenza viruses and flaviviruses[107,108]. Understanding the mechanisms regulating immune imprinting by viruses will inform our evaluation of viral evolution and population immunity, treatment of viral disease, and design of broadly protective vaccines[109,110]. Our study gives experimental evidence for the possible influence of previous seasonal coronavirus infection on the outcome of SARS-CoV-2 exposure. Although it has recently been shown that mice can be infected with select seasonal coronaviruses[111],

our present study expands on this work by showing the effect of a previous exposure to seasonal coronaviruses on SARS-CoV-2 outcomes. To complete the picture of coronavirus imprinting, these studies were done in comparison to a re-exposure with SARS-CoV-2 VOCs. Previous studies focusing on seasonal coronavirus imprinting have been done in humans but have not effectively stratified the contributions of specific seasonal coronaviruses to pre-existing immunity as immunity acquired from the seasonal coronaviruses OC43, NL63, 229E, and HKU1 were analyzed together. In contrast, our study evaluated specific imprinting-challenge combinations of seasonal coronaviruses and SARS-CoV-2 variants and ultimately allowed the stratification of the contribution of a specific previous coronavirus infection to the protection against a subsequent different coronavirus exposure. The results are important for calculating the antigenic distance and immune recall dynamics between coronaviruses to determine mechanisms of cross-protection. Understanding the immunological interactions between different but related viral antigen combinations will be useful for designing pan-coronavirus vaccines and avoiding OAS.

## Methods
### Ethics statement
Ethical approval was obtained for the animal work done in this study. All work was conducted in accordance with the Canadian Council of Animal Care (CCAC) guidelines, Animal Use Protocol (AUP) number 20200019 by the University Animal Care Committee (UACC) Animal Research Ethics Board at the University of Saskatchewan. Hamster procedures were performed under 5% isoflurane anesthesia. All work with Risk Group 3 (RG3) pathogens was performed in the Vaccine and Infectious Disease Organization's (VIDO) Containment Level 3 (CL3) facility (InterVac) in Saskatoon, Saskatchewan, Canada. All work RG2 pathogens was performed in VIDO's CL2 facilities. All work was approved by the University of Saskatchewan's Biosafety Protocol Approval Committee (BPAC) and was overseen by VIDO's Biosafety Officer.

### SARS-CoV-2 variants and viruses
The SARS-CoV-2 pango lineage B isolate /Canada/ON/VIDO-01/2020 was used as a representative for the prototypic B lineage virus (Ancestral virus) (GISAID−EPI_ISL_425177 [https://gisaid.org/][112]. SARS-CoV-2 variants Alpha (GenBank−OQ781006) and Beta (NCBI BioProject−SAMN24618764 [https://www.ncbi.nlm.nih.gov/biosample/SAMN24618764/]) were obtained from clinical isolates collected at Alberta Health and Roy Romanow Provincial Laboratory, respectively. Delta was obtained as a virus stock from the National Microbiology Laboratory (D. Safronetz) (NCBI BioProject−SAMN24618763 [https://www.ncbi.nlm.nih.gov/biosample/?term=SAMN24618763]). Omicron BA.1 was obtained from a clinical isolate collected at BC CDC (GISAID−EPI_ISL_7370259 [https://gisaid.org/]). All SARS-CoV-2 variants and viruses were cultured in vDMEM (viral DMEM (Dulbecco's Modified Eagle Medium) (*Wisent Bioproducts (Cat # 319-005-CL)*), 2% fetal calf serum (*Wisent Bioproducts (Cat # 090−150)*), 5 mL 100x penicillin (10,000 U/mL)/streptomycin (10,000 μg/mL), and 2 μg/mL TPCK-trypsin on Vero-76 cells. Subsequent assays with SARS-CoV-2 viruses were performed using Vero-76 cells and Vero cells. All work with infectious SARS-CoV-2 virus was performed in the Vaccine and Infectious Disease Organization's (VIDO) Containment Level 3 (CL3) facility (InterVac) in Saskatoon, Saskatchewan, Canada.

### Seasonal coronaviruses
Coronaviruses HCoV-NL63, HCoV-229E, and HCoV-OC43 were utilized as representative viruses for seasonally circulating, common-cold coronaviruses. HCoV-NL63 was obtained from BEI (lot. 70037857), grown and titered in LLC-MK2 cells using vDMEM (viral DMEM (Dulbecco's Modified Eagle Medium) (*Wisent Bioproducts*

(Cat # 319-005-CL)), 2% fetal calf serum (Wisent Bioproducts (Cat # 090–150)) and 5 mL 100x penicillin (10,000 U/mL)/streptomycin (10,000 µg/mL). HCoV-229E was obtained from ATCC (lot 70035459) and grown as well as titered using MRC-5 cells in vDMEM (viral DMEM (Dulbecco's Modified Eagle Medium) (Wisent Bioproducts (Cat # 319-005-CL)), 2% fetal calf serum (Wisent Bioproducts (Cat # 090–150)), 5 mL 100x penicillin (10,000 U/mL)/streptomycin (10,000 µg/mL), and 2 µg/mL TPCK-trypsin). HCoV-OC43 was obtained from ATCC (lot. 70035458), grown and titered HCT-8 cells using vRPMI-1640 (viral RPMI-1640 (Roswell Park Memorial Institute 1640 Medium) (ThermoFisher Cat #A104910), 2% horse serum (Wisent Bioproducts (Cat # 065-250)), 5 mL 100x penicillin (10,000 U/mL)/streptomycin (10,000 µg/mL), and 2 µg/mL TPCK-trypsin).

## Animals, infections, and tissue collection
Male, LVG Golden Syrian hamsters aged 8-weeks were purchased from Charles River Laboratories (Wilmington, USA). A total of 108 animals were used in this study. Only male hamsters were used in the present study due to handling difficulties with female Syrian hamsters, which show aggression to one another, in a CL3 environment. Hamsters were anesthetized with 5% isoflurane for intranasal inoculations. All inoculations were performed with 100 µL of human coronaviruses at $10^5$ TCID$_{50}$ per animal. To investigate acute responses following seasonal coronavirus infection, 18 animals were inoculated per group. Three animals were randomly selected and euthanized for necropsies from each group on days 3, 6, and 9. For the primary inoculation-secondary challenge studies, animals were primary inoculated on day 0 and three animals per group were removed on day 55 to serve as a baseline for secondary inoculations. On day 56, animals were inoculated with their respective secondary viruses and monitored over 14 days. Three animals per group were removed at day 3, 6, and 9 post secondary challenge. To calculate weight loss, temperature, and survival, six animals per group were monitored until day 14 post secondary challenge, as we and others have done previously[35,36]. At necropsy, tissues including nasal turbinates, all right lung lobes (cranial, middle, caudal), accessory lung, mediastinal lymph node, heart, kidney, spleen, liver, and large intestines were collected for virological, immunological, and pathological analysis. The left lung lobe was collected for histopathology while the right lobes were used for viral load and RNA analysis. Weight and temperature were calculated as a percentage of original values from day 0. Blood was collected in BD Vacutainer® EDTA coated blood collection tubes for plasma separation.

## Viral titers
Nasal washes (0.3 mL) were collected post inoculation in anesthetized hamsters. Tissues collected at necropsy were homogenized in serum-free DMEM using a Qiagen TissueLyser. To determine TCID$_{50}$, a 1:10 dilution series of sample was established in the cell-appropriate viral growth media in 96-well plates containing virus-designated cells. When determining if infection was possible (229E, NL63, OC43) the initial sample was undiluted to lower the assay's limit of detection. Samples were assessed in triplicate. Cells were incubated for 1 h at 37 °C. Sample dilutions were removed and replaced with fresh viral media. Cytopathic effect (CPE) was monitored daily for 5 days. On day 5 post inoculation of cells, CPE was recorded indicating the presence of infectious virus. These data were used to calculate TCID$_{50}$/mL using the Reed-Muench method[113].

## Viral neutralization assay
Blood was collected at necropsy and centrifuged to yield plasma. Plasma was heat-inactivated at 56 °C for 30 m and then serially diluted 1:2 in a low serum viral media, in round bottom 96-well plates. Virus was diluted to 100 TCID$_{50}$ per well in viral media and used at a 1:1 ratio to plasma, where 60 µL of plasma is in each well and 60 µL of diluted virus is added. The plasma-virus mixture is incubated at 37 °C for 1 h and then added to cultured virus-designated cells in 96-well plates. Plates are then incubated for 1 h at 37 °C, when the plasma-virus mixture was removed from the cells and replaced with fresh viral media. Plates were monitored daily for CPE and endpoint neutralization titer was based on inhibition of CPE observed on day 5 after cell infection. The endpoint titer is reciprocal of the lowest dilution of plasma that is able to suppress CPE.

## RNA extraction and quantitative real-time PCR
Tissue RNA was extracted using the Qiagen© RNeasy Mini kit cat. # 74106 (Qiagen, Toronto, Canada) according to the manufacturer's instructions. vRNA was extracted from nasal washes using the Qiagen© QIAamp Viral RNA Mini Kit cat. # 52904 (Qiagen). vRNA was quantified by Qiagen© Quanti-fast RT probe master mix (Qiagen) using primer/probe sets specific for human coronavirus genes (Table 1)[114]. All host qRT-PCR was performed in triplicate utilizing SYBR Green PCR Mastermix (BioRad) on cDNA synthesized with iScript Reverse Transcriptase (BioRad)[115] (Table 2). The reactions were performed on a StepOnePlus™ Real-Time PCR System in a 96-well plate (ThermoFisher).

## Histopathology
The tissues collected for histopathology were submersed in formalin for at least 7 days within the CL3 lab. Formalin-fixed tissues were paraffin-embedded, sectioned, slide-mounted, and stained at Prairie Diagnostic Services (Saskatoon, Saskatchewan). Tissue samples were stained with hematoxylin and eosin (H&E). H&E stained tissues were imaged using a Aperio ScanScope XT (Leica Biosystems, Nußloch, Germany). Visualization and analysis were performed by a board-certified pathologist. Lungs were given a score ranging from 0 to 4 corresponding to seven categories. Proportion describes the

**Table 1 | Panel of coronavirus gene primers for viral RNA detection**

| Virus | Target | F/R | Sequence (5'>3') |
|---|---|---|---|
| HCoV-229E | Membrane (M) | F | TTCCGACGTGCTCGAACTTT |
| HCoV-229E | Membrane (M) | R | CCAACACGGTTGTGACAGTGA |
| HCoV-NL63 | Nucleocapsid (N) | F | AGGACCTTAAATTCAGACAACGTTCT |
| HCoV-NL63 | Nucleocapsid (N) | R | GATTACGTTTGCGATTACCAAGACT |
| HCoV-OC43 | Membrane (M) | F | ATGTTAGGCCGATAATTGAGGACTAT |
| HCoV-OC43 | Membrane (M) | R | AATGTAAAGATGGCCGCGTATT |
| SARS-CoV-2 | Envelope (E) Genomic | F | ACAGGTACGTTAATAGTTAATAGCGT |
| SARS-CoV-2 | Envelope (E) Subgenomic | F | CGATCTCTTGTAGATCTGTTCTC |
| SARS-CoV-2 | Envelope (E) | R | ATATTGCAGCAGTACGCACACA |
| SARS-CoV-2 | Envelope (E) | P* | ACACTAGCCATCCTTACTGCGCTTCG |

*P = Probe

**Table 2 | Panel of Syrian hamster primers against host response genes**

| Target | F/R | Sequence (5'>3') | Target | F/R | Sequence (5'>3') |
|--------|-----|------------------|--------|-----|------------------|
| BACT | F | ACTGCCGCATCCTCTTCCT | PRF1 | F | TGATAACGGCTGGGACGATG |
| BACT | R | TCGTTGCCAATGGTGATGAC | PRF1 | R | ACTGCCATGGTTCAGGCTAC |
| IFN-β | F | TTGTGCTTCTCCACTACAGC | GZMB | F | GGCCAAGAGGACTAAGGCTG |
| IFN-β | R | GTGTCTAGATCTGACAACCT | GZMB | R | TGCAAAGTTGCCTTTTGGGG |
| STAT2 | F | AATGCCTTCAGAGTGTACCG | GATA-3 | F | GAAGGCAGGGAGTGTGTGAA |
| STAT2 | R | TGTTCACCGTACTATCCACTTCAT | GATA-3 | R | GTCTGACAGTTCGCACAGGA |
| IRF1 | F | GGCATACAACATGTCTTCACG | IL-4 | F | CCACGGAGAAAGACCTCATCTG |
| IRF1 | R | GCTATGCTTTGCCATGTCAA | IL-4 | R | GGGTCACCTCATGTTGGAAATAAA |
| IRF3 | F | AGACGCTAGGGTAGGAAGGG | IL-5 | F | TCACCGAGCTCTGTTGACAA |
| IRF3 | R | GCCACAGCGCAAATCTTTCA | IL-5 | R | CCACACTTCTCTTTTTGGCG |
| TLR3 | F | CTCCGGACTGAAGCAGACAA | IL-13 | F | AAATGGCGGGTTCTGTGC |
| TLR3 | R | GTACTCCAGAGACAGGTGCC | IL-13 | R | AATATCCTCTGGGTCTTGTAGATGG |
| IRF2 | F | AATGCCTTCAGAGTGTACCG | ROR-γ-T | F | TTGGCCAAAACAGAGGTCCA |
| IRF2 | R | TGTTCACCGTACTATCCACTTCAT | ROR-γ-T | R | TTCCAAGAGTAAGTTGGCCGT |
| STAT1 | F | GCCAACGATGATTCCTTTGC | IL-17 | F | ATGTCCAAACACTGAGGCCAA |
| STAT1 | R | GCTATATTGGTCATCCAGCTGAGA | IL-17 | R | GCGAAGTGGATCTGTTGAGGT |
| CXCL10 | F | GCCATTCATCCACAGTTGACA | IL-22 | F | TTCACCCTTGCAGAAGTGCT |
| CXCL10 | R | CATGGTGCTGACAGTGGAGTCT | IL-22 | R | AGGCTGAGCTGGTTGCTTAG |
| IFN-γ | F | GGCCATCCAGAGGAGCATAG | FoxP3 | F | GGTCTTCGAGGAGCCAGAAGA |
| IFN-γ | R | TTTCTCCATGCTGCTGTTGAA | FoxP3 | R | GCCTTGCCCTTCTCATCCA |
| IL-12 | F | GGCCTTCCCTGGCAGAA | IL-10 | F | GTTGCCAAACCTTATCAGAAATGA |
| IL-12 | R | ATGCTGAAAGCCTGCAGTAGAAT | IL-10 | R | TTCTGGCCCGTGGTTCTCT |
| T-bet | F | ACAAGGGGGCTTCCAACAAT | TGF-β1 | F | TGTGTGCGGCAGCTGTACA |
| T-bet | R | CAGCTGAGTGATCTCGGCAT | TGF-β1 | R | TGGGCTCGTGAATCCACTTC |
| CD3 | F | CTGGCTGCTTTTCTCCCTCG | CD19 | F | GGGAACCAGTCAACACCCTT |
| CD3 | R | AACCATACTTTCGCCGTTCCC | CD19 | R | TTAGCAAGATTCCCAGGGGC |
| IL-2 | F | GTGCACCCACTTCAAGCTCTAA | AID | F | AAGAGGCGTGACAGTGCTAC |
| IL-2 | R | AAGCTCCTGTAAGTCCAGCAGTAAC | AID | R | CAGTCCGAGATGTAGCGGAG |
| IL-21 | F | TCAACTGATGTGAAAGGAGC | BCL6 | F | GCCATCTCCCACTCAGTAGC |
| IL-21 | R | ATCTTGTGGAGCTGGCAG | BCL6 | R | TCCGTTCCTTGGGAAAAGGG |
| CXCR5 | F | ATGGCCTGGTGTTATGTGGG | IL-1β | F | GGCTGATGCTCCCATTCG |
| CXCR5 | R | GTCACTAAGATGGCCACCCG | IL-1β | R | CACGAGGCATTTCTGTTGTTCA |
| CD4 | F | GTGTCAAGTGCCGACACCA | TNF | F | GGAGTGGCTGAGCCATCGT |
| CD4 | R | AGAAGAGTTGTGAGGGTCCCA | TNF | R | AGCTGGTTGTCTTTGAGAGACATG |
| CD8 | F | GTGCTGTTGACCATTTCGCA | IL-6 | F | CCTGAAAGCACTTGAAGAATTCC |
| CD8 | R | TTCCCGGGTCGAAGAGAGAT | IL-6 | R | GGTATGCTAAGGCACAGCACACT |

percentage of parenchyma affected where 1 = <25%; 2 = 26–50%; 3 = 51–75% 4 = 76–100%. Inflammation indicated the density of the inflammatory infiltrate in affected areas. Pneumonia details the extent of hypertrophy and/or hyperplasia of alveolar pneumocytes. Bronchioles scores corresponds epithelial lesions in small bronchioles (extending into terminal bronchioles). Hemorrhage rates intra-alveolar hemorrhage. Bronchus focuses on intrapulmonary portion of a large bronchus and its larger tributaries including infiltration of inflammatory cells in wall. Overall gives a summary score taking all factors into account. 0, absent (no lesion); 1, slight or questionable; 2, clearly present, but not conspicuously so; 3, marked; 4, severe.

## Viral identity analysis
The raw sequences of viral strains were collected from GISAID. We applied the bioinformatic technique BLOcks SUbstitution Matrices (BLOSUM) for comparative analysis. BLOSUM is a substitution matrix for protein sequence alignments, which is based on maximum identity percentage of the aligned protein sequences[116]. Viral sequences were imported into Geneious Prime (2022.2.2) as a FASTA file and sequences were selected to perform pairwise alignment using Geneious

Alignment. Parameters were set to global alignment with free end gaps at cost matrix BLOSUM90 and a gap open penalty and extension penalty at 12 and 3 respectively.

## Epitope prediction
For MHC-I Prediction, S-protein reference sequences for a SARS-CoV-2 pango B lineage (NCBI accession no. YP_009724390.1), Beta (GISAID accession no. EPI_ISL_864621 [https://gisaid.org/]), Omicron (GISAID accession no. EPI_ISL_6699766.1 [https://gisaid.org/]), NL63 (NCBI accession no. YP_003767.1), and OC43 (NCBI accession no. YP_009555241.1) were screened for predicted MHC-I compatible peptides through the NetMHCpan EL 4.1 program[117]. Alleles were selected based on HLA:A and HLA:B variants representative of roughly 97% of the general population from the following list: HLA-A01:01, HLA-A26:01, HLA-A32:01, HLA-A02:01, HLA-A02:03, HLA-A02:06, HLA-A68:02, HLA-A23:01, HLA-A24:02, HLA-A03:01,HLA-A11:01, HLA-A30:01, HLA-A31:01, HLA-A33:01, HLA-A68:01, HLA-B40:01, HLA-B44:02, HLA-B44:03, HLA-B57:01, HLA-B58:01, HLA-B15:01, HLA-B07:02, HLA-B35:01, HLA-B51:01, HLA-B53:01, and HLA-B08:01[118]. Predicted MHC-I compatible antigens

9-10 residues in length were filtered by percentile binding score and compared between variants. Peptides were filtered for those only displaying a percentile score below or equal to 0.05.

MHC-II binding was predicted using NetMHCII 2.3 nn-align[119] against a similar list of 27 alleles representative of the world population majority as follows: HLA-DRB1*01:01, HLA-DRB1*03:01, HLA-DRB1*04:01, HLA-DRB1*04:05, HLA-DRB1*07:01, HLA-DRB1*08:02, HLA-DRB1*09:01, HLA-DRB1*11:01, HLA-DRB1*12:01, HLA-DRB1*13:02, HLA-DRB1*15:01, HLA-DRB3*01:01, HLA-DRB3*02:02, HLA-DRB4*01:01, HLA-DRB5*01:01, HLA-DQA1*05:01/DQB1*02:01, HLA-DQA1*05:01/DQB1*03:01, HLA-DQA1*03:01/DQB1*03:02, HLA-DQA1*04:01/DQB1*04:02, HLA-DQA1*01:01/DQB1*05:01, HLA-DQA1*01:02/DQB1*06:02, HLA-DPA1*02:01/DPB1*01:01, HLA-DPA1*01:03/DPB1*02:01 HLA-DPA1*01:03/DPB1*04:01, HLA-DPA1*03:01/DPB1*04:02, HLA-DPA1*02:01/DPB1*05:01, and HLA-DPA1*02:01/DPB1*14:01[120]. Peptides were selected between 13-17 residues in length and predicted epitopes were screened for binding affinity scores representative of the top 0.2 percentile. Predicted MHC compatible peptides were compared to ancestral SARS CoV-2 using the Epitope conservancy Analysis tool (IEDB, http://tools.iedb.org/conservancy/)[121].

### Statistical analysis
Unpaired, unequal variance, two-tail Student's *t*-test or one-way ANOVAs were conducted using GraphPad Prism8 (San Diego, USA). A *p* value of $\leq 0.05$ was considered statistically significant.

### Reporting summary
Further information on research design is available in the Nature Portfolio Reporting Summary linked to this article.

## Data availability
All accession codes, publicly available datasets and data generated and analyzed during this study are included in this published article and its Supplementary Information files. Source data are provided with this paper.

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

## Acknowledgements

We would like to sincerely thank the BC CDC (British Columbia Centre for Disease Control) for contributing virus isolates. Funding has been provided by the Canadian Institutes for Heath Research (CIHR) Operating Grant: Emerging COVID-19 Research Gaps (GA2-177714) (A.A.K.) and Priorities for Vaccines and the CIHR and Coalition for Epidemic Preparedness (CEPI) Leadership Award in Vaccine Research (CLV 179494-2022) (A.A.K.), CIHR Project Grant: Pandemic Preparedness and Health Emergencies Research (PPE – 185823) CIHR grants OV5-170349 (D.F.), VRI-173022 (D.F.), VS1-175531 (D.F.), and the CIHR-funded Coronavirus Variants Rapid Response Network (CoVaRR-Net) (D.F.). This article is published with the permission of the Director of VIDO, manuscript no. 1012. VIDO receives operational funding from the Government of Saskatchewan through Innovation Saskatchewan and the Ministry of Agriculture and from the Canada Foundation for Innovation through the Major Science Initiatives for its CL3 Facility.

## Author contributions

A.A.K. conceived and designed the experiments. M.E.F., E.B.J., A.Y., A.S., C.L.S., B.K.M., B.M.T., R.B., M.B.R., C.S., and A.A.K. performed the experiments. M.E.F., E.B.J., A.Y., A.S., K.J.L., J.D., M.B.R., C.S., and A.A.K. analyzed the data. J.L., V.G., D.F., D.M.S., C.S., and A.A.K. contributed materials/analysis tools. M.E.F., D.M.S., and A.A.K. wrote the paper.

## Competing interests

The authors declare no competing interests.

## Additional information

[1]Vaccine and Infectious Disease Organization VIDO, University of Saskatchewan, Saskatoon, SK, Canada. [2]Department of Biochemistry, Microbiology, and Immunology, University of Saskatchewan, Saskatoon, SK, Canada. [3]BC Centre for Disease Control, Immunization Programs and Vaccine Preventable Diseases Service, Vancouver, BC, Canada. [4]University of British Columbia, School of Population and Public Health, Vancouver, BC, Canada. [5]Department of Psychiatry, Queen's University, Kingston, ON, Canada. [6]Queen's Genomics Lab at Ongwanada (Q-GLO), Ongwanada Resource Centre, Kingston, ON, Canada. ✉e-mail: alyson.kelvin@usask.ca

