## [Peer Review File · Nature Communications]

REVIEWER COMMENTS

Reviewer #1 (Remarks to the Author):

Francis and colleagues present a set of hamster studies probing infection outcomes of consecutive alpha and beta-coronavirus infections comparing serological, virological, histopathological and expressional outcomes. We see a potential strength of the study in the assessment of the interference between heterologous primary and secondary infections (pages 16-21). Thus obtained signatures hint towards intriguing interactions. However, due to very small group sizes, and limited breadth and vigor in analysis key findings appear preliminary and remain non-conclusive.

The abstract has a by far too long narrative and my need focus on key hypothesis, results and message. The entire text may profit from focus and more concise language. Several terms are used somewhat floppy such as “seasonal imprinting” whereas “primary exposure to seasonal coronaviruses” is meant.

Missing line numbering makes any point-by-point assessment unnecessarily tedious.

Specific comments and concerns

Abstract

- “coronaviruses at various distances” is misleading terminology; “distinct CoV of various evolutionary distances” or “more or less distantly related CoVs”
- „using the Syrian hamster model“ – using a hamster model (there is not one that has been used for that before), or “using Syrian hamsters as infection model“
- “hamsters were imprinted” is not correct; hamsters were may become imprinted by exposure, thus imprinting may be a result of exposure but may hardly be deliberately induced experimentally. The authors should discriminate more carefully between their experimental hypothesis and actual experimental interventions.
- “original SARS-CoV-2 Wuhan virus“ – prototypic SARS-CoV-2 Wuhan strain virus
- „mounted antigenic sin-like antibody responses“ -- reword antigenic sin-like
- „imprinting-challenge regimens“ – heterologous infection regimens
- „suggesting Omicron to be more antigenically distant from Wuhan than the Beta variant“ – Omicron is more distant, thus rather confirming yet here likely functionally
- “inflammation and antiviral response suppression“ – in this syntax it is not clear whether inflammation is also suppressed

- „These results are important for calculating the antigenic distance between coronaviruses and for determining positive or negative cross-reactive signatures.“ – one means of assessing distances complementary to phylogenetic, antigenic cartography etc.

- „matrix of protection“ – not a generally used term, thus not unambiguous what meant here

Introduction

- „Immunological imprinting, previously known only for its negative effects“ – not true, imprinting of innate immunity has been shown to be able to contribute to protection

- “imprinting has been most associated with sequential influenza virus infection“ – not most associated, but like “best studied”

- “past spillover coronaviruses“ – not sure if all this evolutionary and epidemiological info is needed for understand; also very long complicated sentences in first paragraph of intro

- „immunologically dominant surface Spike (S) protein“ – S may be dominant protective antigen in vaccines, but not sure that is also immunologically dominant in natural infection; infections seems to mount B- and T-cell immunity against many other viral proteins such as N; to our knowledge dominance of S over other antigens has never been demonstrated

- “emergence of more immune evasive variants“ – emergence of variants evading (or escaping) immunity

- “represented more closely related virus exposures“ – more importantly also represent what happened recently to large parts of the human population; subsequent exposure by prototype and/or beta, then omicron

Results

- „available for this project“ – availability should not be relevant, rather rationale

- „Previously published studies of seasonal coronavirus infection mainly leveraged the mouse model while studies of SARS-CoV-2 and variant infections by our group and others have used Syrian hamsters to assess pathogenesis, host responses, and vaccines 30-36. Since Syrian hamsters are outbred, we hypothesized these animals to be a more representative model of human genetic variability for our study. To give insight into the relatedness of the human coronaviruses and the potential of using hamsters as a single animal species for this work, we first conducted analysis on the S protein sequences for the coronaviruses of interest and the corresponding Syrian hamster sequences of the specific viral host cell receptors (Fig. 1A).“ – please condense

- “This viral sequence analysis suggested that the seasonal coronaviruses were antigenically distant to the SARS-CoV-2 Wuhan and VOC viruses which were more similar to each other.“ – trivial, please condense

- “We were not able to acquire HKU1 for this study, therefore the susceptibility to HKU1 was not investigated.“ – delete

- “Three groups of Syrian hamsters were intranasally inoculated“ – please provide group size n, age and sex

- “Although live virus was not identified in the lungs, histopathological analysis of lung tissue indicated immune cell infiltration and mild pathology.” – please highlight some characteristic pathological changes in the histopathological pictures in Figure-1D,E e.g. by arrows or other symbols. Reporting lung weight may not be sufficient. Adding histological scores may enhance the data.
- „To understand the immune responses in the respiratory tissues of NL63 and OC43 inoculated animals, we assessed host gene expression of select immune genes via qPCR. In the nasal turbinates (Extended Data Fig. 3)“ – these key data on differential expression profiles should be considered as main figure instead of supplementary
- “we went on to determine the impact and mechanisms of seasonal coronavirus imprinting on a SARS-CoV-2 Wuhan secondary challenge 26-28 – these cited references are reporting conflicting results; please mention that this is to be experimentally assessed and unveiled
- “were imprinted with” – see above; were infected with to possibly imprint
- “nadir” – not commonly used
- „psc“ – abbreviation not introduced before
- „remained statistically above“ – somewhat weird expression
- „Histopathological analysis of the lungs after secondary Wuhan challenge showed characteristic interstitial pneumonia, hemorrhaging, and mononuclear cell infiltration into the alveolar space peaking on day 6 psc of the mock-Wuhan group which was similar to our previous study of Wuhan infected younger Syrian hamsters 34. Both the NL63-Wuhan and OC43-Wuhan groups had significantly less evidence of interstitial pneumonia in the lungs following Wuhan challenge throughout the time course.“ – these data seem not to be presented
- „during Wuhan challenge, the NL63- Wuhan group had a significant increase or a back-boosting effect of NL63 neutralizing antibodies following Wuhan exposure with titers increasing above 1:320 by day 14 psc“ – very small group size and relatively small changes may not allow to draw these conclusions
- „However, both seasonal virus imprinted groups had trends of higher neutralizing antibody titers against the antigenically distant variants of Delta, Beta, and Omicron compared to the mock primary group.“ – too small groups and no proper statistics done to support this conclusion
- „Imprinting with Wuhan protects against secondary challenge with antigenically divergent SARS- CoV-2 variants (...) These results suggest Beta and Wuhan to be antigenically closer to each other than to Omicron and that the heterologous exposure of Wuhan followed by Beta may be an optimal antigenic distance to induce a greater breadth of antibody cross-reactivity.“ – these data have been presented by others before and may not be result of imprinting but need to be rather called cross-protection amongst SARS-CoV-2 variants.
- Entire section “Host response analysis during secondary challenge indicates immune regulation is based on antigenic distance“ is very interesting and may be considered key of the entire study, however proper cluster analysis of groups and statistics lacking in Figure 6 are needed.

- „More predicted T cell epitopes on OC43 shared with SARS-CoV-2 Wuhan“ – this section is irrelevant for the hamster model as no MHC-I nor II predictions are possibly in lack of data for this model. Modelling humans HLA-restricted peptides does not support finding in the previous key paragraph.

The discussion should be compacted and focus on the proposed interference between primary and secondary infections.

Reviewer #2 (Remarks to the Author):

In their manuscript titled, “Signatures of Original Antigenic Sin or protection following coronavirus imprinting in Syrian hamsters challenged with SARS-CoV-2 or variants”, Francis et al characterize the impact of prior infection with seasonal coronaviruses on infection, pathogenesis, and immune responses to heterologous zoonotic coronavirus secondary infections. In doing so, they identify differential impacts of NL63 and OC43 primary infections on virological, disease, and immune-response outcomes to CoV2 secondary infections. Interestingly, the differential disease outcomes appeared to correlate with back-boosting of NL63 neutralizing antibody responses and T-cell recall responses to OC43, with the former resulting in poor control of, and the latter improving host response to, secondary exposure. The methods and study designs were appropriate, and the results support the major conclusions. A few minor notes described below.

Figure 3 describes the neutralizing antibody responses to both the seasonal and zoonotic coronaviruses, with panel A providing an overview of the kinetics of response, only to the primary infecting virus, throughout the entire study while panel B provides a snapshot of the days 9 and 14 post-secondary infection, only to the zoonotic secondary infecting virus and its variants. To gain a better understanding of the recall responses to the secondary infection, it would improve the interpretation of the study if the authors could include either a reproduction of panel Ai and Aii against Wuhan virus or just the day 0 post-secondary challenge in panel B so that fold-change nAb responses can be calculated for NL63, OC43, and Wuhan. This would allow readers to interpret the relative focusing of the recall responses between the primary and secondary infecting viruses. While panel Ai clearly shows back-boosting of NL63 responses, it is unclear how that boosting response compares to the Wuhan nAb response in those same animals since only day 9 and 14 responses are shown in panel B.

Also in figure 3, the homologous wuhan-wuhan sera show remarkable breadth of nAb responses against VoCs with only a 2-fold drop in nAb titers against Omicron and even a potential increased titer against Beta. This is surprising given observations by others showing much larger differences in heterologous nAb titers between Wuhan and Omicron/Beta VoCs. A discussion on this with some potential explanations would be helpful.

Finally, this study only evaluated recall responses over a short period following primary infection (56 days) with minimal waning of primary nAb responses prior to secondary infection. A discussion on this limitation and impact on conclusions is warranted.

Reviewer #3 (Remarks to the Author):

In this manuscript, Francis and colleagues use the Syrian hamster model to assess the relative role previous infection with seasonal coronaviruses contributes to subsequent infection and host response elicitation with SARS-CoV-2 virus. Authors conduct primary challenge experiments with several seasonal coronaviruses, followed by rechallenge with an early SARS-CoV-2 virus, or primary challenge with the early Wuhan virus followed by rechallenge with different variants of concern. A range of virological and host response parameters were conducted. Collectively the authors find that primary infection with seasonal coronaviruses did not confer robust protection against SARS-CoV-2 virus, but do identify subtle differences in host responses using several parameters, providing information towards understanding the diversity of responses possible when assessing correlates of protection towards coronaviruses moving forward. The manuscript is well-written, well-referenced, logically organized, and presents a clear rationale for experiments performed. There are some areas that would nonetheless benefit from additional clarity and justification as discussed below.

Comments:

-Page 4, 2nd paragraph of introduction, it would be helpful to the reader if you mentioned what relative percentage of the population had been infected with seasonal coronaviruses prior to the start of the SARS-CoV-2 pandemic, to better contextualize why preclinical models with preexisting antibody to seasonal CoV is important to emulate.

-Page 7, it is understandable (and not a detriment to the study) that HKU1 could not be included. However, can the authors speculate (here or in the discussion) how this virus might have behaved in the analyses included (e.g. based on the spike protein identity analysis shown in Fig 1) if this virus would have behaved relative to the other seasonal coronaviruses examined?

-In rechallenge experiments, animals were rested from day 14 p.i. to day 56 p.i., at which point secondary challenge took place. Why was this interval chosen and do the authors think that a longer duration of resting between challenges would have modulated the rechallenge results?

-Page 12, please specify which text maps to Figure 2D and 2E; it is currently unclear how the lung weight differences presented in Figure 2E align with the text as currently written in this section.

-Page 14, authors state that both seasonal virus imprinted groups had “trends of higher neutralizing antibody titers” against SARS-CoV-2 variants. However, the data shown in Figure 3B (to my eye) shows generally comparable antibody titers among all mock, NL63, and OC43 primary-challenged animals; are the authors drawing from additional data to make this statement, or comparing mean values that can be reported which reflect this trend? As currently presented in the graphs, without additional supporting information, I think the higher “trends” alluded to here are a bit spurious.

-page 16/Figure 5, the current graphs as presented make it a bit challenging for the reader to compare trends present from specific groups of animals that are present between different assessments of neutralizing antibodies (the graph are designed for, and effectively show, augmentation of neutralizing antibody titers against a single variant over time in different treated groups of animals). If the authors want to state, as they do in the text, that specific animal groups displayed trends across multiple assays shown in Figure 5, an additional panel in this figure (e.g. where the legend is currently present), showing comparative levels of one timepoint assessed for the presence of different neutralizing antibodies between different groups, would make it much easier for the reader to quickly see the data the authors are alluding to in making this statement.

-Figure 6, please specify what fold change is in relation to (mock animals?) and what blue in the scale bar refers to (values with a negative fold-change relative to mock?). Also please specify what normalization (if any) took place in these specimens; authors show in earlier figures that lungs had different weights so I presume some sort of normalization to a housekeeping gene was conducted?

-Figure 1C, the coloration between live virus and qPCR in these graphs is very, very subtle; recommend a different differentiation strategy (e.g. colors with greater separation, or filled vs open symbols) to show this difference. Also please clarify why the LOD for these data is different than in figure 2C.

REVIEWER COMMENTS

Reviewer #1

Remarks to the Author:

Francis and colleagues present a set of hamster studies probing infection outcomes of consecutive alpha and beta-coronavirus infections comparing serological, virological, histopathological and expressional outcomes. We see a potential strength of the study in the assessment of the interference between heterologous primary and secondary infections (pages 16-21). Thus obtained signatures hint towards intriguing interactions. However, due to very small group sizes, and limited breadth and vigor in analysis key findings appear preliminary and remain non-conclusive.

The abstract has a by far too long narrative and my need focus on key hypothesis, results and message. The entire text may profit from focus and more concise language. Several terms are used somewhat floppy such as “seasonal imprinting” whereas “primary exposure to seasonal coronaviruses” is meant. Missing line numbering makes any point-by-point assessment unnecessarily tedious.

Response to Reviewer #1’s Remarks: We thank Reviewer #1 for their critical review of our manuscript. The abstract has been condensed as suggested.

Comments:

Abstract:

Reviewer 1, Comment 1: “coronaviruses at various distances” is misleading terminology; “distinct CoV of various evolutionary distances” or “more or less distantly related CoVs”

Response to Reviewer 1, Comment 1: Thank you for this correction. We have replaced “coronaviruses at various distances” with “distinct coronaviruses of various evolutionary distances” on line 32 and 33.

Reviewer 1, Comment 2: „using the Syrian hamster model“ – using a hamster model (there is not one that has been used for that before), or “using Syrian hamsters as infection model“

Response to Reviewer 1, Comment 2: We have corrected this to “using Syrian hamsters as infection model” as the reviewer suggested, as seen on line 33.

Reviewer 1, Comment 3: “hamsters were imprinted” is not correct; hamsters were may become imprinted by exposure, thus imprinting may be a result of exposure but may hardly be deliberately induced experimentally. The authors should discriminate more carefully between their experimental hypothesis and actual experimental interventions.

Response to Reviewer 1, Comment 3: We appreciate this discrepancy and have corrected “imprinted” to “inoculated” or “inoculated with the intent to imprint” throughout the manuscript, including line 34.

Reviewer 1, Comment 4: “original SARS-CoV-2 Wuhan virus” – prototypic SARS-CoV-2 Wuhan strain virus

Response to Reviewer 1, Comment 4: Thank you for this clarification. We now refer to the virus as “prototypic, Ancestral SARS-CoV-2 pango B lineage Wuhan strain virus” on

lines 35 and 36. We have also changed the nomenclature in all references to “Wuhan” to “Ancestral” throughout the manuscript and figures.

Reviewer 1, Comment 5: „mounted antigenic sin-like antibody responses“ -- reword antigenic sin-like

Response to Reviewer 1, Comment 5: We have reworded “mounted antigenic sin-like antibody responses” at the reviewer’s suggestion to “mounted antibody responses with signatures of original antigenic sin” on lines 43 and 44.

Reviewer 1, Comment 6: „imprinting-challenge regimens“ – heterologous infection regimens

Response to Reviewer 1, Comment 6: In place of “imprinting-challenge regimens” we have used “heterologous infection regimens” on line 45.

Reviewer 1, Comment 7: „suggesting Omicron to be more antigenically distant from Wuhan than the Beta variant“ – Omicron is more distant, thus rather confirming yet here likely functionally

Response to Reviewer 1, Comment 7: Thank you for this clarification. On line 46, we have replaced “suggesting Omicron to be more antigenically distant” to “supporting Omicron to be more antigenically distant”.

Reviewer 1, Comment 8: “inflammation and antiviral response suppression” – in this syntax it is not clear whether inflammation is also suppressed

Response to Reviewer 1, Comment 8: We apologize for being unclear. We have replaced “inflammation and antiviral response suppression” to “increased inflammation as well as suppression of the antiviral response were signatures in antigenically distant viral challenge” on line 49 and 50.

Reviewer 1, Comment 9: „These results are important for calculating the antigenic distance between coronaviruses and for determining positive or negative cross-reactive signatures.“ – one means of assessing distances complementary to phylogenetic, antigenic cartography etc.

Response to Reviewer 1, Comment 9: The sentence has now been modified to read “These results are important for assessing antigenic distances between coronaviruses complementary to phylogenetics and antigenic cartography” on lines 52 to 53.

Reviewer 1, Comment 10: „matrix of protection“ – not a generally used term, thus not unambiguous what meant here

Response to Reviewer 1, Comment 10: We have removed this sentence.

Introduction:

Reviewer 1, Comment 11: „Immunological imprinting, previously known only for its negative effects“ – not true, imprinting of innate immunity has been shown to be able to contribute to protection

Response to Reviewer 1, Comment 11: This sentence has been modified to "Immunological imprinting, previously described as '*Original Antigenic Sin*', is the powerful mark, positive or negative, that a first viral infection makes on the host's immune system and response to subsequent antigenically related viral exposures." On lines 57 to 59.

Reviewer 1, Comment 12: "imprinting has been most associated with sequential influenza virus infection" – not most associated, but like "best studied"

Response to Reviewer 1, Comment 12: We appreciate this discrepancy. We have corrected "imprinting has been most associated with sequential influenza virus infection" to "imprinting has been best studied in the context of sequential influenza virus infection" on lines 61 and 62.

Reviewer 1, Comment 13: "past spillover coronaviruses" – not sure if all this evolutionary and epidemiological info is needed for understand; also very long complicated sentences in first paragraph of intro

Response to Reviewer 1, Comment 13: The sentence has now been modified to "Human coronaviruses (HCoV) belong to either the alphacoronavirus or betacoronavirus genera suggesting these genera to be of human concern" on lines 72 to 74.

Reviewer 1, Comment 14: „immunologically dominant surface Spike (S) protein“ – S may be dominant protective antigen in vaccines, but not sure that is also immunologically dominant in natural infection; infections seems to mount B- and T-cell immunity against many other viral proteins such as N; to our knowledge dominance of S over other antigens has never been demonstrated

Response to Reviewer 1, Comment 14: We appreciate this comment. We have added three citations demonstrating S as the immunologically dominant protein in the context of infection to line 74.

Reviewer 1, Comment 15: "emergence of more immune evasive variants" – emergence of variants evading (or escaping) immunity

Response to Reviewer 1, Comment 15: At the reviewer's suggestion, we have modified the sentence to "emergence of variants of concern (VOCs) evading immunity" on lines 93 and 94.

Reviewer 1, Comment 16: "represented more closely related virus exposures" – more importantly also represent what happened recently to large parts of the human population; subsequent exposure by prototype and/or beta, then omicron

Response to Reviewer 1, Comment 16: We have modified this sentence to include the sentiment that these exposures were relevant to a large portion of the current global population. The sentence now reads "The results were then compared to imprinting-challenge studies performed with Ancestral SARS-CoV-2 virus and the VOCs, Beta and

Omicron, as these combinations represented more closely related virus exposures as well as recent exposures affecting large populations of people.” On lines 106 to 109.

Results:

Reviewer 1, Comment 17: „available for this project“ – availability should not be relevant, rather rationale

Response to Reviewer 1, Comment 17: We appreciate this comment. We have removed HKU1 availability from the text.

Reviewer 1, Comment 18: „Previously published studies of seasonal coronavirus infection mainly leveraged the mouse model while studies of SARS-CoV-2 and variant infections by our group and others have use Syrian hamsters to assess pathogenesis, host responses, and vaccines 30-36. Since Syrian hamsters are outbred, we hypothesized these animals to be a more representative model of human genetic variability for our study. To give insight into the relatedness of the human coronaviruses and the potential of using hamsters as a single animal species for this work, we first conducted analysis on the S protein sequences for the coronaviruses of interest and the corresponding Syrian hamster sequences of the specific viral host cell receptors (Fig. 1A).“ – please condense

Response to Reviewer 1, Comment 18: Thank you for the comment. We have condensed the section to now read, “Previously published studies of seasonal coronavirus infection mainly leveraged the mouse model; however, we chose to use Syrian hamsters for our study of human coronaviruses due the utility it has shown in SARS-CoV-2 pathogenesis studies.” On lines 120 to 123.

Reviewer 1, Comment 19: “This viral sequence analysis suggested that the seasonal coronaviruses were antigenically distant to the SARS-CoV-2 Wuhan and VOC viruses which were more similar to each other.“ – trivial, please condense

Response to Reviewer 1, Comment 19: We have now revised this sentence to “This viral sequence analysis suggested that the seasonal coronaviruses were antigenically distant to the SARS-CoV-2 Ancestral strain while VOCs were more antigenically similar” on line 127 to 129.

Reviewer 1, Comment 20: “We were not able to acquire HKU1 for this study, therefore the susceptibility to HKU1 was not investigated. – delete

Response to Reviewer 1, Comment 20: “We were not able to acquire HKU1 for this study, therefore the susceptibility to HKU1 was not investigated.” has been deleted from the text.

Reviewer 1, Comment 21: “Three groups of Syrian hamsters were intranasally inoculated“ – please provide group size n, age and sex

Response to Reviewer 1, Comment 21: We apologize for not including this information. “Male hamsters, aged 8-weeks were utilized. A total of 12 animals were used in each group (three animals sampled on days 3 and 6 post inoculation and six animals sampled

on day 14 post inoculation).” Has been added to lines 143 to 145; “A total of 18 male Syrian hamsters were used in each group (three animals sampled on days 0, 3, 6 and 9 post secondary challenge and six animals sampled on day 14 post secondary challenge).” Similar text has been added in appropriate places throughout the manuscript.

Reviewer 1, Comment 22: “Although live virus was not identified in the lungs, histopathological analysis of lung tissue indicated immune cell infiltration and mild pathology.” – please highlight some characteristic pathological changes in the histopathological pictures in Figure-1D,E e.g. by arrows or other symbols. Reporting lung weight may not be sufficient. Adding histological scores may enhance the data.

Response to Reviewer 1, Comment 22: We appreciate this suggestion. To support our histological data, we have included pathological scoring in the new Extended Data Fig. 2, 7 and 9.

The description of this data to support Fig. 1D and E now reads “Although live virus was not identified in the lungs, histopathological analysis of lung tissue indicated immune cell infiltration and mild pathology. Mononuclear cells infiltrated the alveoli with associated inflammation on day 6 and 14 pi following NL63 inoculation and small clusters of macrophage-like cells were observed lasting until day 14 (Fig. 1D). The lungs of NL63 inoculated animals scored for inflammation peaking at an average of 1.3/4 on day 6 and somewhat resolving to .33/4 on average on day 14 (Extended Data Fig. 2). The OC43 inoculated animals had significant mononuclear cell infiltration (mainly macrophage-like cells and neutrophils) and hemorrhaging on day 6 pi. Edema was also noted around a group of blood vessels in one of three animals on day 14 post OC43 inoculations (Fig. 1D). The same OC43 inoculated animal scored a 2/4 on lung inflammation with less than 25% of the lung being affected (Proportion score of 1) (Extended Data Fig. 2). Not surprisingly the lungs from the 229E inoculated animals, which did not show evidence of active viral infection even in the upper respiratory tract, retained a similar architecture compared to control uninfected tissue (Fig. 1D) and scored 0/4 on lung histology (Extended Data Fig. 2). The SARS-CoV-2 Ancestral virus inoculated animals displayed evidence of bronchopneumonia, significant peribronchiolar thickening, hemorrhaging, interstitial pneumonia, and mononuclear cell infiltration on day 6 pi (Fig. 1D) as we have previously reported⁴¹. Ancestral virus inoculated animals scored on all histology parameters, with over 50% of the lung being affected on day 6 and 14, and highest scoring in any category being on day 6 with an average score of 3 in bronchiole, corresponding to marked epithelial lesions (Extended Data Fig. 2).” On lines 172 to 191.

Reviewer 1, Comment 23: „To understand the immune responses in the respiratory tissues of NL63 and OC43 inoculated animals, we assessed host gene expression of select immune genes via qPCR. In the nasal turbinates (Extended Data Fig. 3)“ – these key data on differential expression profiles should be considered as main figure instead of supplementary

Response to Reviewer 1, Comment 23: Thank you for this suggestion. To address this, we have made an additional main text figure, now Fig. 2 to include a succinct representation

of the data from Extended Data Fig. 3 and 4 (Now Extended Data Fig. 4 and 5), which remain in the extended data, as represented on lines 206 to 219.

Reviewer 1, Comment 24: “we went on to determine the impact and mechanisms of seasonal coronavirus imprinting on a SARS-CoV-2 Wuhan secondary challenge 26-28 – these cited references are reporting conflicting results; please mention that this is to be experimentally assessed and unveiled

Response to Reviewer 1, Comment 24: Thank you for pointing this out. We have addressed that these references are conflicting with the addition of “given the conflicting results reported in human clinical studies^{21,34,35}” on lines 229 and 230.

Reviewer 1, Comment 25: “were imprinted with” – see above; were infected with to possibly imprint

Response to Reviewer 1, Comment 25: “were imprinted with” has been adjusted to “were intranasally inoculated with” on lines 232 and 233.

Reviewer 1, Comment 26: “nadir” – not commonly used

Response to Reviewer 1, Comment 26: “nadir” was replaced with “the lowest point” on line 248.

Reviewer 1, Comment 27: „psc“ – abbreviation not introduced before

Response to Reviewer 1, Comment 27: The abbreviation is described in its first use on line 242.

Reviewer 1, Comment 28: „remained statistically above“ – somewhat weird expression

Response to Reviewer 1, Comment 28: We appreciate this note. We have replaced “remained statistically above” with “remained statistically increased relative to” on line 255.

Reviewer 1, Comment 29: „Histopathological analysis of the lungs after secondary Wuhan challenge showed characteristic interstitial pneumonia, hemorrhaging, and mononuclear cell infiltration into the alveolar space peaking on day 6 psc of the mock-Wuhan group which was similar to our previous study of Wuhan infected younger Syrian hamsters 34. Both the NL63-Wuhan and OC43-Wuhan groups had significantly less evidence of interstitial pneumonia in the lungs following Wuhan challenge throughout the time course.“ – these data seem not to be presented

Response to Reviewer 1, Comment 29: We apologize for this confusion. To support our histological data, we have included pathological scoring in the new Extended Data Fig. 7 and refined the conclusions made from the histology datasets. Fig. 2D (Now Fig. 3D) is described below.

The description of this data now reads “Histopathological analysis of the lungs after secondary Ancestral virus challenge showed characteristic interstitial pneumonia,

hemorrhaging, and mononuclear cell infiltration into the alveolar space peaking on day 6 psc of the Mock-Ancestral virus group (Fig. 3D) which was similar to our previous study of Ancestral virus infected younger Syrian hamsters⁴¹. Histology scoring correspondingly peaked on day 6 when there was an overall severity score of 3/4 on average and scoring of 1 or greater in all animals for inflammation, pneumonia, and bronchioles (Extended Data Fig. 7). NL63-Ancestral animals but not OC43-Ancestral animals had an increased pneumonia scoring in the lungs following Ancestral challenge throughout on day 6 (Extended Data Fig. 7). Of note, the NL63 primary group had marked accumulation of red blood cells in the epithelium, blood vessels, and alveolar space on day 3 psc (Hemorrhage score of 2/4). Small clusters of mononuclear cells were evident in the parenchymal interstitium and, to a lesser extent, in the alveoli on day 3 and 6psc (Fig. 3D, Extended Data Fig. 7). In general, the OC43 imprinted animals had lungs with less evidence of immune cell infiltration and pathology compared to NL63 imprinted animals, with lower inflammation scoring and proportion of the lung affected in OC43 imprinted animals on days 6 and 14 psc (Extended data Fig 7), although epithelial sloughing in the bronchioles was evident on day 3 psc and mild interstitial pneumonia was present on day 6 psc. Areas of pneumocyte hypertrophy were also observed lasting until day 14 psc (Fig. 3D). The Ancestral-Ancestral virus control group had little evidence of alveolar wall thickening and the alveolar space remained clear; however, there was significant presence of red blood cells in the alveolar wall and mononuclear cell infiltrates in the bronchiolar walls on day 3 psc. Some hyperplasia of alveolar pneumocytes were also noted on day 3 and 6 psc (Fig. 3D). No animal in the Ancestral-Ancestral group scored over a 2 on any lung histology severity marker, with an overall peak on day 3 at 1.3/4 (Extended Data Fig. 7).” On lines 274 to 296

Reviewer 1, Comment 30: „during Wuhan challenge, the NL63- Wuhan group had a significant increase or a back-boosting effect of NL63 neutralizing antibodies following Wuhan exposure with titers increasing above 1:320 by day 14 psc“ – very small group size and relatively small changes may not allow to draw these conclusions

Response to Reviewer 1, Comment 30: Thank you for this comment. We have clarified that despite the small n in Fig. 4Ai (current figure lineup), these differences were found to be significantly different: “The day 14 psc NL63 neutralizing antibody titers were significantly increased compared to day 14 ppi (1:160), as well as just prior to the secondary inoculation, on day 55 ppi, (1:80) (Fig. 4Ai).” lines 318 to 320.

Reviewer 1, Comment 31: „However, both seasonal virus imprinted groups had trends of higher neutralizing antibody titers against the antigenically distant variants of Delta, Beta, and Omicron compared to the mock primary group.“ – too small groups and no proper statistics done to support this conclusion

Response to Reviewer 1, Comment 31: We appreciate the comment. We have addressed that the trends observed were non-significant, “both seasonal virus imprinted groups had non-significant trends of higher neutralizing antibody titers against the antigenically distant variants of Delta, Beta, and Omicron compared to the mock primary group” on lines 341 to 343.

Reviewer 1, Comment 32: „Imprinting with Wuhan protects against secondary challenge with antigenically divergent SARS- CoV-2 variants (...) These results suggest Beta and Wuhan to be antigenically closer to each other than to Omicron and that the heterologous exposure of Wuhan followed by Beta may be an optimal antigenic distance to induce a greater breadth of antibody cross-reactivity.“ – these data have been presented by others before and may not be result of imprinting but need to be rather called cross-protection amongst SARS-CoV-2 variants.

Response to Reviewer 1, Comment 32: We have modified the text to reflect the phenomenon of cross-protection. In lines 345 and 346 the text is now “Imprinting with ancestral SARS-CoV-2 induces cross-protection against secondary challenge with antigenically divergent variants”. Also, in lines 398 and 399 the text now reads “These results suggest cross-protective responses where Beta and Ancestral SARS-CoV-2 are antigenically closer to each other than to Omicron”.

Reviewer 1, Comment 33: Entire section “Host response analysis during secondary challenge indicates immune regulation is based on antigenic distance“ is very interesting and may be considered key of the entire study, however proper cluster analysis of groups and statistics lacking in Figure 6 are needed.

Response to Reviewer 1, Comment 33: We appreciate this suggestion. To accommodate this, we have included the fully detailed bar graph representation of this data in newly added Extended Data Fig. 10, 11 and 13, described on lines 1504 to 1544.

Reviewer 1, Comment 34: „More predicted T cell epitopes on OC43 shared with SARS-CoV-2 Wuhan“ – this section is irrelevant for the hamster model as no MHC-I nor II predictions are possibly in lack of data for this model. Modelling humans HLA-restricted peptides does not support finding in the previous key paragraph. The discussion should be compacted and focus on the proposed interference between primary and secondary infections.

Response to Reviewer 1, Comment 34: Thank you for pointing out the caveat that we should have identified previously. We have now shortened the T cell analysis section and highlighted the caveat of using the human MHC molecules for the analysis. The modified section is from lines 513 to 541.

Reviewer #2

Remarks to the Author:

In their manuscript titled, “Signatures of Original Antigenic Sin or protection following coronavirus imprinting in Syrian hamsters challenged with SARS-CoV-2 or variants”, Francis et al characterize the impact of prior infection with seasonal coronaviruses on infection, pathogenesis, and immune responses to heterologous zoonotic coronavirus secondary infections. In doing so, they identify differential impacts of NL63 and OC43 primary infections on virological, disease, and immune-response outcomes to CoV2 secondary infections. Interestingly, the differential disease outcomes appeared to correlate with back-boosting of NL63 neutralizing antibody responses and T-cell recall responses to OC43, with the former resulting in poor control of, and the latter improving host response to, secondary exposure. The methods and study designs were appropriate, and the results support the major conclusions. A few minor notes described below.

Response to Reviewer #2's Remarks: We thank Reviewer #2 for their comments as they have improved the interpretation and discussion of the manuscript. The comments are addressed as described below.

Comments:

Reviewer 2, Comment 1 : Figure 3 describes the neutralizing antibody responses to both the seasonal and zoonotic coronaviruses, with panel A providing an overview of the kinetics of response, only to the primary infecting virus, throughout the entire study while panel B provides a snapshot of the days 9 and 14 post-secondary infection, only to the zoonotic secondary infecting virus and its variants. To gain a better understanding of the recall responses to the secondary infection, it would improve the interpretation of the study if the authors could include either a reproduction of panel Ai and Aii against Wuhan virus or just the day 0 post-secondary challenge in panel B so that fold-change nAb responses can be calculated for NL63, OC43, and Wuhan. This would allow readers to interpret the relative focusing of the recall responses between the primary and secondary infecting viruses. While panel Ai clearly shows back-boosting of NL63 responses, it is unclear how that boosting response compares to the Wuhan nAb response in those same animals since only day 9 and 14 responses are shown in panel B.

Response to Reviewer 2, Comment 1: This is a very important point, and we apologize the data was confusing. To address this, we have expanded upon Fig. 3B (Now Fig. 4B) to include all time points post secondary challenge, including day 0 (labelled 'pre'). This change has been described in the figure legend outlined on lines 1316 to 1318. We originally left the other time points out because there were no detectable neutralizing antibodies, but we see that led to confusion. We hope the inclusion of the other time points improves the interpretation of the data.

Reviewer 2, Comment 2 : Also in figure 3, the homologous wuhan-wuhan sera show remarkable breadth of nAb responses against VoCs with only a 2-fold drop in nAb titers against Omicron and even a potential increased titer against Beta. This is surprising given observations by others showing much larger differences in heterologous nAb titers between Wuhan and Omicron/Beta VoCs. A discussion on this with some potential explanations would be helpful.

Response to Reviewer 2, Comment 2: Thank you for the comment. We have added some points in the discussion on possibilities for differences among studies to explain Fig. 3B (Now Fig. 4B). The following lines were added to the discussion at lines 710 to 715, "Additionally, we found a high level of cross-neutralizing antibodies among SARS-CoV-2 Ancestral virus and the VOCs. Previous studies have also investigated cross-protection and cross-neutralization among SARS-CoV-2 VOCs with some studies showing high levels of cross-neutralization and others with lower levels^{101,105}. Differences in cross-neutralization or protection compared to the findings from our study could be due to lower infectious doses at inoculation or time between primary infection and secondary challenge or blood collection".

Reviewer 2, Comment 3 : Finally, this study only evaluated recall responses over a short period following primary infection (56 days) with minimal waning of primary nAb responses prior to secondary infection. A discussion on this limitation and impact on conclusions is warranted.

Response to Reviewer 2, Comment 3: Thank you for asking about this limitation. To address this, we have added a discussion on the rationale on using this timeline, in addition to hypotheses as to what would happen at alternative secondary infection timepoints on lines 598 to 617.

The following was included in the Discussion: “Our study design had a relatively short recovery time of 56 days between primary infection and secondary challenge. We leveraged this design as it was long enough for allow innate responses to return to baseline and adaptive responses to begin contracting while also being short enough for data acquisition and completion of our many groups. Due to this design our conclusions can only reflect responses in this timeframe. Since the acute infection phase of COVID-19 in humans is considered to be 2-4 weeks^{77,78}, to understand the development of long-lived memory cells and the recall response upon secondary exposure, a longer recovery time should be studied in the future. In our study we observed a decrease in antibodies against NL63 and OC43 over the recovery period which was not observed for the SARS-CoV-2 ancestral virus infected group. It would be interesting to investigate longer time periods for the seasonal coronaviruses as we expect that antibody levels would continue to decline, as has been seen in humans^{17,79,80}. Given this, upon exposure to SARS-CoV-2 at a later timepoint when antibody responses are waning, it is possible that less of an antigenic sin-like response towards NL63 would be induced. Several studies have now investigated long-term memory development against SARS-CoV-2 in humans⁸¹⁻⁸⁵. Dan et al. demonstrated that 8-months after infection, B cell responses were relatively maintained in patient sera, while the CD4 and CD8 T cell responses decreased over time⁸¹ suggesting that if we extended our recovery period after SARS-CoV-2 exposure the humoral response may be sustained while the decrease of CD4 cells over time may affect the germinal center response through a T follicular helper cells dependent mechanism⁸⁶⁻⁸⁸. More work is needed to understand how cross-reactivity among coronaviruses may change over time.”

Reviewer #3

Remarks to the Author:

In this manuscript, Francis and colleagues use the Syrian hamster model to assess the relative role previous infection with seasonal coronaviruses contributes to subsequent infection and host response elicitation with SARS-CoV-2 virus. Authors conduct primary challenge experiments with several seasonal coronaviruses, followed by rechallenge with an early SARS-CoV-2 virus, or primary challenge with the early Wuhan virus followed by rechallenge with different variants of concern. A range of virological and host response parameters were conducted. Collectively the authors find that primary infection with seasonal coronaviruses did not confer robust protection against SARS-CoV-2 virus, but do identify subtle differences in host responses using several parameters, providing information towards understanding the diversity of responses possible when assessing correlates of protection towards coronaviruses moving forward. The manuscript is well-written, well-referenced, logically organized, and presents a clear rationale for experiments performed. There are some areas that would nonetheless benefit from additional clarity and justification as discussed below.

Response to Reviewer #3’s remarks: We thank reviewer 3 for their interest and appreciation of the study. Additionally, we thank the reviewer for suggestions that will improve the manuscript. Comments are addressed below.

Comments:

Reviewer 3, Comment 1 : Page 4, 2nd paragraph of introduction, it would be helpful to the reader if you mentioned what relative percentage of the population had been infected with seasonal coronaviruses prior to the start of the SARS-CoV-2 pandemic, to better contextualize why preclinical models with preexisting antibody to seasonal CoV is important to emulate.

Response to Reviewer 3, Comment 1: We appreciate this point and think it is very important to include. Thank you! To better contextualize why preclinical models including the seasonal coronaviruses are important we added context as to what percentage of the population have been exposed to seasonal coronaviruses and what the reinfection dynamics. This description is on lines 80 to 87.

The section included in the introduction is as follows: "It is estimated that adults experience 2-3 colds annually while children can have even more common cold virus events each year¹⁸. Additionally, several human studies suggest that immunity to common colds is short-lasting, and reinfection can occur yearly.¹⁷ While several different viruses can cause the common cold, coronaviruses are estimated to be the cause of 15-30% of common colds¹⁹. Therefore, although almost all people are thought to have been infected with cold viruses at some point in their lives, 30% to 60% of people will have been infected with a seasonal coronavirus within the last year.^{20-22.}"

Reviewer 3, Comment 2 : Page 7, it is understandable (and not a detriment to the study) that HKU1 could not be included. However, can the authors speculate (here or in the discussion) how this virus might have behaved in the analyses included (e.g. based on the spike protein identity analysis shown in Fig 1) if this virus would have behaved relative to the other seasonal coronaviruses examined?

Response to Reviewer 3, Comment 2: Thank you for this question. Given the spike identity analysis in Fig. 1A as well as characteristics of HKU1 we have speculated on how HKU1 imprinting may affect a secondary exposure to SARS-CoV-2 on lines 637 to 644.

The section in the discussion now reads "In our present study we did not include analysis of HKU1 due to availability of the virus. However, since HKU1 is also a betacoronavirus and is second only to OC43 in terms of spike amino acid identity when compared to SARS-CoV-2, we anticipate that animals imprinted with HKU1 would have similar outcomes and potential cross-reactive T cell responses as we have found in the OC43 imprinted animals. This may be further supported by the fact that HKU1 uses the same entry receptor as OC43, 9-O-acetylated sialic acid."

Reviewer 3, Comment 3 : In rechallenge experiments, animals were rested from day 14 p.i. to day 56 p.i., at which point secondary challenge took place. Why was this interval chosen and do the authors think that a longer duration of resting between challenges would have modulated the rechallenge results?

Response to Reviewer 3, Comment 3: Thank you for this question. We feel the conversation to be very important for the field (and wish we could have compared many time points!). To address this question (which was also posed by Reviewer #2) as well as

hypothesize what a longer duration of resting between imprinting and challenge, we have expanded upon our discussion on lines 598 to 617.

The inserted paragraph is as follows:

The following was included in the Discussion: “Our study design had a relatively short recovery time of 56 days between primary infection and secondary challenge. We leveraged this design as it was long enough for allow innate responses to return to baseline and adaptive responses to begin contracting while also being short enough for data acquisition and completion of our many groups. Due to this design our conclusions can only reflect responses in this timeframe. Since the acute infection phase of COVID-19 in humans is considered to be 2-4 weeks^{77,78}, to understand the development of long-lived memory cells and the recall response upon secondary exposure, a longer recovery time should be studied in the future. In our study we observed a decrease in antibodies against NL63 and OC43 over the recovery period which was not observed for the SARS-CoV-2 ancestral virus infected group. It would be interesting to investigate longer time periods for the seasonal coronaviruses as we expect that antibody levels would continue to decline, as has been seen in humans^{17,79,80}. Given this, upon exposure to SARS-CoV-2 at a later timepoint when antibody responses are waning, it is possible that less of an antigenic sin-like response towards NL63 would be induced. Several studies have now investigated long-term memory development against SARS-CoV-2 in humans⁸¹⁻⁸⁵. Dan et al. demonstrated that 8-months after infection, B cell responses were relatively maintained in patient sera, while the CD4 and CD8 T cell responses decreased over time⁸¹ suggesting that if we extended our recovery period after SARS-CoV-2 exposure the humoral response may be sustained while the decrease of CD4 cells over time may affect the germinal center response through a T follicular helper cells dependent mechanism⁸⁶⁻⁸⁸. More work is needed to understand how cross-reactivity among coronaviruses may change over time.”

Reviewer 3, Comment 4 : Page 12, please specify which text maps to Figure 2D and 2E; it is currently unclear how the lung weight differences presented in Figure 2E align with the text as currently written in this section.

Response to Reviewer 3, Comment 4: We apologize for the confusion. We have expanded our description of Fig. 2E (Now Fig. 3E) and specified those results distinct from the Fig.2D (now Fig. 3D) results on lines 296 to 302.

The text now reads: “Some hyperplasia of alveolar pneumocytes were also noted on day 3 and 6 psc (Fig. 3D). No animal in the Ancestral-Ancestral group scored over a 2 on any lung histology severity marker, with an overall peak in pathological scoring on day 3 at 1.3/4 (Extended Data Fig. 7). The damage to the lung as noted in the histopathology in the Mock-Ancestral group was accompanied by an increase in lung weight which peaked on day 6 post inoculation at 5% of animal body weight (Fig. 3E). Ancestral-Ancestral animals had a significant decrease in lung weight compared to Mock- Ancestral animals on all days psc, whereas the seasonal coronavirus imprinted groups had significant decrease in lung weight compared to Mock-imprinted animals on days 3 and 14, but not day 4 psc (Fig. 3E).

Reviewer 3, Comment 5 : Page 14, authors state that both seasonal virus imprinted groups had “trends of higher neutralizing antibody titers” against SARS-CoV-2 variants. However, the data shown in Figure 3B (to my eye) shows generally comparable antibody titers among all mock, NL63, and OC43 primary-challenged animals; are the authors drawing from additional data to make this statement, or comparing mean values that can be reported which reflect this trend? As currently presented in the graphs, without additional supporting information, I think the higher “trends” alluded to here are a bit spurious.

Response to Reviewer 3, Comment 5: We apologize for the lack of clarity in this figure. To improve the interpretation of this data, we have included the results from days 0, 3 and 6 post-secondary inoculation, in addition to the day 9 and 14 data that was originally presented in Fig. 3B (Now Fig 4B). We have also specified this trend is non-significant and applies only to Delta and Beta, as indicated on lines 341 to 343. We hope the data is better understood now. The text now reads “both seasonal virus imprinted groups had non-significant trends of higher neutralizing antibody titers against the antigenically distant variants of Delta and Beta, compared to the mock imprinted group.”

Reviewer 3, Comment 6 : page 16/Figure 5, the current graphs as presented make it a bit challenging for the reader to compare trends present from specific groups of animals that are present between different assessments of neutralizing antibodies (the graph are designed for, and effectively show, augmentation of neutralizing antibody titers against a single variant over time in different treated groups of animals). If the authors want to state, as they do in the text, that specific animal groups displayed trends across multiple assays shown in Figure 5, an additional panel in this figure (e.g. where the legend is currently present), showing comparative levels of one timepoint assessed for the presence of different neutralizing antibodies between different groups, would make it much easier for the reader to quickly see the data the authors are alluding to in making this statement.

Response to Reviewer 3, Comment 6: Thank you for this comment. We have now expanded Fig. 5 (Now Fig. 6) to include a graph with just day 14 post-secondary inoculation, allowing to statistically show differences between these groups (Fig.6B).

The description of Fig. 6B is on lines 391 to 399 and reads “The Ancestral-Omicron group had neutralizing titers on day 14 psc with geometric means >1:320, >1:320, >1:320, and >1:640, for the Ancestral strain, and Beta, Alpha, and Delta variants, respectively (Fig. 6B). The Ancestral-Omicron group had titers of 1:1280 against the Omicron virus itself. Ancestral-Ancestral and Ancestral-Beta groups were above 1:640 for all variants with the exception of the Omicron neutralization assay which were lower (Fig. 6B). Interestingly, the Ancestral-Beta group had a higher median than the Ancestral-Ancestral group for the neutralization of Beta, Omicron, Alpha, and Delta viruses (Fig. 6B).”

Reviewer 3, Comment 7 : Figure 6, please specify what fold change is in relation to (mock animals?) and what blue in the scale bar refers to (values with a negative fold-change relative to mock?). Also please specify what normalization (if any) took place in these specimens; authors show in earlier figures that lungs had different weights, so I presume some sort of normalization to a housekeeping gene was conducted?

Response to Reviewer 3, Comment 7: To clarify these points, we have added a description of fold change calculations, the scale bar as well as normalization to a housekeeping gene in the figure legend for Fig. 6 (Now Fig. 7), on lines 1361 to 1364.

The text now read “Fold-change was calculated via $\Delta\Delta C_t$ against baseline (Day 0) with BACT as the housekeeping gene. The legend depicts fold change in which upregulation (greater than 1) is represented ranging in from white (=1) to red. Any downregulation (less than 1) is represented in blue.”

Reviewer 3, Comment 8 : Figure 1C, the coloration between live virus and qPCR in these graphs is very, very subtle; recommend a different differentiation strategy (e.g., colors with greater separation, or filled vs open symbols) to show this difference. Also please clarify why the LOD for these data is different than in Figure 2C.

Response to Reviewer 3, Comment 8: We apologize for this oversight. We have updated these colours in Fig. 1C so that the data can more easily be differentiated between groups. The LOD in these assays was lower than other assays as we decreased the dilution series by starting with an undiluted sample rather than a 1:10 dilution. This strategy was taken so that we could increase the possibility of detecting lower amounts of virus that may be in the sample. This description has been added to the methods for clarification in lines 810 and 811.

REVIEWER COMMENTS

Reviewer #1 (Remarks to the Author):

The authors make a great effort to adapt their narrative according our suggestion; to clarify and correct possibly misleading expressions. For me my main criticism regarding the lack of clear experimental evidence for 'Original Antigen Sin' remains. Fig-3 may be key here. In brief, NL63: no effect (neither on viral loads nor on pathology; similar to mock); OC43: minimal effect on lung viral loads; none on pathology); effect only seen when animals are second time exposed to (any) SARS-CoV-2 isolate. This cannot be called Antigenic Sin this is adaptive immunity. Some level of cross-protection between ancestral SARS-CoV-2 and VoCs is well documented before. In my opinion the authors should consider focussing on the OC43-SARS-CoV-2 interaction to substantiate their hypothesis.

Minor Comments:

- 1-The Abstract may be by far too long.
- 2-Old Comment 14: The references provided show that S contain distinct B- and T-cells epitopes that dominate S-specific immune responses; not that responses to S are dominating during natural infection.
- 3-I suggest to write "ancestral" with a small letter
- 4-Old Comment 21: group sizes appear too small to draw quantitative conclusions.
- 5-Old Comment 23: The authors are encouraged to improve their analysis of the DEG analysis e.g. clustering, statistics, etc. to go beyond plain reporting of primary results.
- 6-Line 274-276: It is misleading to call mock-infected (thus previously non-infected) animals that are infected for the first time with SARS-COV-2 exposed to a "secondary ancestral virus challenge".
- 7-Line 345: This describes consecutive infection by two SARS-CoV-2 viruses. "Imprinting" is wrong (as it implies already the to be proven hypothesis); what is actually meant is "prior infection". Nevertheless, cross protection from prior SAR-CoV-2 is now long standing knowledge and thus an almost trivial finding.
- 8- Old Comment 34: This extensive narrative immunoinformatic discovery of T-cell epitopes is misleading because related to the human model and not the hamster model as used to generate the original results in the MS. I suggest to cut down, make this narrative much less suggestive, and/or move to Supplementary.

Reviewer #2 (Remarks to the Author):

Thank you for responding to my requests. I found the responses to be appropriate and the revised version of the manuscript is easier to interpret (at least for me).

Reviewer #3 (Remarks to the Author):

Authors have addressed all comments raised during peer review; no further comments.

REVIEWER COMMENTS:

Reviewer #1

Remarks to the Author:

The authors make a great effort to adapt their narrative according our suggestion; to clarify and correct possibly misleading expressions. For me my main criticism regarding the lack of clear experimental evidence for 'Original Antigen Sin' remains. Fig-3 may be key here. In brief, NL63: no effect (neither on viral loads nor on pathology; similar to mock); OC43: minimal effect on lung viral loads; none on pathology); effect only seen when animals are second time exposed to (any) SARS-CoV-2 isolate. This cannot be called Antigenic Sin this is adaptive immunity. Some level of cross-protection between ancestral SARS-CoV-2 and VoCs is well documented before. In my opinion the authors should consider focussing on the OC43-SARS-CoV-2 interaction to substantiate their hypothesis.

Response to Reviewer #1's Remarks: Thank you for the notes on the revised manuscript. We have removed all reference to Original Antigenic Sin in the title as well as the introduction of the manuscript. All notes on Original Antigenic Sin are in the Discussion section and mainly present in the final paragraph.

Comments:

Reviewer 1, Comment 1: The Abstract may be by far too long.

Response to Reviewer 1, Comment 1: Thank you for the attention to making a better version of the Abstract. We have now further reduced the abstract to 199 words.

Reviewer 1, Comment 2: Old Comment 14: The references provided show that S contain distinct B- and T-cells epitopes that dominate S-specific immune responses; not that responses to S are dominating during natural infection.

Response to Reviewer 1, Comment 2: We appreciate this distinction. We have updated lines 51 and 52 to read "The Spike (S) protein on the viral envelope contains distinct B- and T-cells epitopes that dominate S-specific immune responses⁹⁻¹¹".

Reviewer 1, Comment 3: I suggest to write "ancestral" with a small letter.

Response to Reviewer 1, Comment 3: We have updated the manuscript to have "ancestral" written with a lowercase 'a' in all cases with the exception of animal group names such as "Mock-Ancestral".

Reviewer 1, Comment 4: Old Comment 21: group sizes appear too small to draw quantitative conclusions.

Response to Reviewer 1, Comment 4: The group sizes were designed according to standard in the field. It is also important to acknowledge the realistic capacity of housing hamsters in CL3 containment as well as working in CL3 containment. Hamsters are ten times larger than mice and typical animal numbers for mice cannot be accommodated for in hamster studies. All efforts were made to follow standards in the field of SARS-CoV-2 infection in hamsters which were achieved and present in the manuscript. Each group had between 6 and 12 animals per group in the analysis. In total, our study used 181 hamsters.

Reviewer 1, Comment 5: Old Comment 23: The authors are encouraged to improve their analysis of the DEG analysis e.g. clustering, statistics, etc. to go beyond plain reporting of primary results.

Response to Reviewer 1, Comment 5: Thank you for this suggestion. We are a bit unclear about what specific analysis the review is asking for. Given that this is qPCR data and not RNA sequencing data, further analysis such as clustering is most appropriately done with next generation sequencing transcriptome data which was not this analysis.

Reviewer 1, Comment 6: Line 274-276: It is misleading to call mock-infected (thus previously non-infected) animals that are infected for the first time with SARS-COV-2 exposed to a "secondary ancestral virus challenge".

Response to Reviewer 1, Comment 6: Thank you for this point. To clarify, we have adjusted lines 223 to 224 to now say "...significantly less than the weight loss during the study day 56 challenge period of the Mock-Ancestral control animals".

Reviewer 1, Comment 7: Line 345: This describes consecutive infection by two SARS-CoV-2 viruses. "Imprinting" is wrong (as it implies already the to be proven hypothesis); what is actually meant is "prior infection". Nevertheless, cross protection from prior SAR-CoV-2 is now long standing knowledge and thus an almost trivial finding.

Response to Reviewer 1, Comment 7: Line 323: The title of this section has been changed to *Previous infection with ancestral SARS-CoV-2 induces cross-protection against secondary challenge with antigenically divergent variants*

Reviewer 1, Comment 8: Old Comment 34: This extensive narrative immunoinformatic discovery of T-cell epitopes is misleading because related to the human model and not the hamster model as used to generate the original results in the MS. I suggest to cut down, make this narrative much less suggestive, and/or move to Supplementary.

Response to Reviewer 1, Comment 8: Line 482. Thank you for the note and apologize that this section was not cut down enough the first time. We have now significantly shortened

the T cell analysis section and combined the section with the gene expression. Figure 8 has now been moved to Supplementary and is now Extended Data Fig. 14.

Reviewer #2

Remarks to the Author:

Thank you for responding to my requests. I found the responses to be appropriate and the revised version of the manuscript is easier to interpret (at least for me).

Response to Reviewer #2's Remarks: We are very grateful for the comments and are happy our changes had the desired outcome of a more interpretable manuscript.

Reviewer #3

Remarks to the Author:

Authors have addressed all comments raised during peer review; no further comments.

Response to Reviewer #3's Remarks: We would like to sincerely thank Reviewer 3 for their analysis of this manuscript and their constructive comments which ultimately lead to an improved piece of work.